# Distributed-Order Fractional Graph Operating Network

**Kai Zhao**[1]*, **Xuhao Li**[2]*, **Qiyu Kang**[1]†, **Feng Ji**[1],
**Qinxu Ding**[3], **Yanan Zhao**[1], **Wenfei Liang**[1], **Wee Peng Tay**[1]

[1]Nanyang Technological University, [2]Anhui University, [3] Singapore University of Social Sciences

## Abstract

We introduce the Distributed-order fRActional Graph Operating Network (DRAGON), a novel continuous Graph Neural Network (GNN) framework that incorporates distributed-order fractional calculus. Unlike traditional continuous GNNs that utilize integer-order or single fractional-order differential equations, DRAGON uses a learnable probability distribution over a range of real numbers for the derivative orders. By allowing a flexible and learnable superposition of multiple derivative orders, our framework captures complex graph feature updating dynamics beyond the reach of conventional models. We provide a comprehensive interpretation of our framework's capability to capture intricate dynamics through the lens of a non-Markovian graph random walk with node feature updating driven by an anomalous diffusion process over the graph. Furthermore, to highlight the versatility of the DRAGON framework, we conduct empirical evaluations across a range of graph learning tasks. The results consistently demonstrate superior performance when compared to traditional continuous GNN models. The implementation code is available at `https://github.com/zknus/NeurIPS-2024-DRAGON`.

## 1 Introduction

Graph Neural Networks (GNNs) have been developed to handle graph-structured data, which is prevalent in domains such as social networks [1], traffic networks [2], and molecular structures [3]. The fundamental principle of GNNs is to learn representations of nodes or entire graphs that encompass both the attributes of individual nodes and the topology of their connections. This objective is accomplished through a method known as message passing or information propagation, whereby each node aggregates information from its neighbors and possibly itself, over multiple iterations or layers [4]. Recent developments in the GNN landscape have increasingly embraced the principles of continuous dynamical systems for information propagation, as discussed in [5]. This trend is exemplified in works such as CGNN [6], GRAND [7], GRAND++ [8], GraphCON [9], Beltrami [10], GREAD [11], CDE [12], and HANG [13], which employ ordinary or partial differential equations (ODEs/PDEs) on graphs for feature aggregation. Within these continuous GNN models, the differential operator $\frac{\mathrm{d}^\alpha}{\mathrm{d}t^\alpha}$ is typically constrained to integer values of $\alpha$, primarily 1 or 2.

Two directions have been proposed recently based on the aforementioned continuous GNN models to enhance their capabilities. One approach is TDE-GNN [14], which proposes to learn higher integer-order temporal dependencies for continuous GNN models. The other approach is FROND [15], which incorporates graph neural Fractional-order Differential Equations (FDEs), extending the conventional integer-order derivative $\frac{\mathrm{d}^\alpha}{\mathrm{d}t^\alpha}$ to encompass a positive real number $\alpha$. This adaptation not only bolsters the model's efficacy but also enhances its adversarial robustness by varying the value of $\alpha$ [16].

---

*First two authors contributed equally to this work.
†Correspondence to: Qiyu Kang <kang0080@e.ntu.edu.sg>.

38th Conference on Neural Information Processing Systems (NeurIPS 2024).

TDE-GNN, however, is limited to utilizing integer-order ODEs and does not account for the non-local memory effects inherent in fractional-order differential operators. These operators [17] have been developed to overcome the limitations of their traditional integer-order counterparts when modeling complex real-world dynamics. The key difference between fractional and integer operators can be grasped from a microscopic random walk perspective as shown in [15, 18]. For instance, traditional integer-order diffusion PDEs, which model diffusive transport in homogeneous porous media, typically ignore the waiting times between particle movements. However, these models struggle when applied to solute diffusion in heterogeneous porous media, prompting the introduction of fractional-order operators to better handle these complexities [19, 20]. In fractional scenarios, particles may remain at their current position, delaying jumps to subsequent locations with fading waiting times and leading to a non-Markovian process. In contrast, traditional integer-order differential equations are typically used to model Markovian movement of particles, as the derivative $\frac{\mathrm{d}f(t)}{\mathrm{d}t} = \lim_{\Delta t \to 0} \frac{f(t+\Delta t)-f(t)}{\Delta t}$ captures the local rate of function changes. On the other hand, although FROND utilizes a fractional-order $\alpha$ and demonstrates performance improvement, its capacity for feature updating dynamics remains constrained by limited temporal dependencies with a single $\alpha$. Moreover, the optimized performance of FROND is achieved through extensive fine-tuning of $\alpha$ across various graph datasets. Observations from Fig. 2 indicate that performance can fluctuate significantly as the value of the fractional order varies from 0 to 1.

The distributed-order fractional differential operator has gained recognition in fractional calculus for its capacity to model complex dynamics that traditional differential equations with integer or single fractional orders cannot sufficiently capture [21]. Inspired by this advancement, we introduce a novel continuous GNN framework named the *Distributed-order fRActional Graph Operating Network (DRAGON)*, which extends beyond existing frameworks like TDE-GNN and FROND. Rather than designating a single, constant $\alpha$ with extensive fine-tuning, DRAGON employs a learnable measure $\mu$ over a range $[a, b]$ for $\alpha$. The foundation of our framework is the distributed-order fractional differential operator [22]:

$$\int_a^b D^\alpha f(t)\, \mathrm{d}\mu(\alpha), \tag{1}$$

which can be perceived as the limiting case of $\sum_i w(\alpha_i)D^{\alpha_i}f(t)$, a weighted summation over derivatives of multiple orders with weight $w(\cdot)$ (we employ this more common notation $D^\alpha$ instead of $\mathrm{d}^\alpha/\mathrm{d}t^\alpha$ henceforth). Notably, unlike TDE-GNN, which restricts $\alpha_i$ to integer values, DRAGON allows for a continuous range of values, significantly broadening its application scope and flexibility in modeling. This operator also addresses the limitations of the single fractional-order operator $D^\alpha$ employed in FROND, which still has a restricted capacity to model the intricacies of feature updating dynamics. From the perspective of a random walk in a diffusion process, a single $D^\alpha$ dictates that the waiting time between particle jumps follows a fixed power-law distribution $\propto t^{-\alpha-1}$ for $0 < \alpha < 1$. In contrast, DRAGON adopts a more flexible approach, enabling a broader range of waiting times across multiple temporal scales. In this paper, we demonstrate the efficacy of the DRAGON framework in modeling more intricate non-Markovian node feature updating dynamics in graph-based data. We provide evidence that DRAGON can approximate any given waiting time probability distribution pertinent to graph random walks, thus showcasing its advanced capability in capturing complex feature dynamics.

**Main contributions.** Our objective is to develop a general continuous GNN framework that enhances flexibility in graph feature updating dynamics. Our key contributions are summarized as follows:

- We propose a generalized continuous GNN framework that incorporates distributed-order fractional derivatives, extending previous continuous GNN models into a unified approach. Specifically, our framework treats these models as special cases with $\mu(\alpha)$ taking a single positive real value for [7, 8, 11, 13, 15] or multiple integer values [9, 14]. Our approach facilitates flexible and learnable node feature updating dynamics stemming from the superposition of dynamics across various derivative orders.
- From a theoretical standpoint, we present the non-Markovian graph random walk with flexible waiting time for DRAGON, presuming that the feature updating dynamics adhere to a diffusion principle. This exposition elucidates the rationale behind the flexible feature updating dynamics.
- Through empirical assessments, we test the DRAGON-enhanced versions of several prominent continuous GNN models. Our findings consistently demonstrate their outperformance. This under-

scores the DRAGON framework's potential as an augmentation to amplify the effectiveness of a range of continuous GNN models.

## 2 Preliminaries and Related Work

This paper focuses on developing a new GNN framework centered around distributed-order fractional dynamic processes. In this section, we provide a concise introduction to the key concepts in fractional calculus. Throughout the paper, we adopt certain standard assumptions to ensure problem well-posedness. For instance, the well-definedness of integrations, the existence and uniqueness of the differential equation solution [23, 24], and the allowance for interchange between summation and limit via the monotone or dominated convergence theorem [25] are all assumed.

### 2.1 Fractional Derivative

The single fractional-order operator $D^\alpha$ in the distributed-order fractional operator in (1) can assume various definitions. In this study, we start off with the *Marchaud–Weyl* fractional derivative $_\mathrm{M}D^\alpha$, recognized for its efficacy in elucidating the fading memory phenomena [26–28], which we will discuss further in Sections 2.1.1 and 3.2.

**Remark 1.** *However, in practical engineering implementations, the* Caputo *fractional derivative $_\mathrm{C}D^\alpha$ is more commonly utilized [15, 17]. Due to space limitations, the introduction of Caputo's derivative is deferred to the Appendix B and will be subsequently employed in Section 3.3 to solve DRAGON. The Marchaud–Weyl and Caputo definitions are equivalent under certain constraints [17, 29].*

For any $\alpha \in (0, 1)$, the Marchaud–Weyl $\alpha$-order derivative of a function $f$, defined over the real line, at a specified point $t$ is defined as [29]:

$$_\mathrm{M}D^\alpha f(t) = \frac{\alpha}{\Gamma(1-\alpha)} \int_0^\infty \frac{f(t) - f(t-\tau)}{\tau^{1+\alpha}} \, \mathrm{d}\tau, \tag{2}$$

where $\Gamma(\cdot)$ is the Gamma function. For sufficiently smooth functions, according to [29], we have

$$\lim_{\alpha \to 1^-} {}_\mathrm{M}D^\alpha f(t) = \frac{\mathrm{d}f(t)}{\mathrm{d}t} = \lim_{\Delta t \to 0} \frac{f(t + \Delta t) - f(t)}{\Delta t}. \tag{3}$$

It is evident from (2) that the Marchaud–Weyl fractional derivative is a nonlocal operator and accounts for the past values of $f$ within the $(-\infty, t)$ range, indicative of its memory effect. In terms of probability, the related non-Markovian processes for fractional systems are characterized by state evolution that depends not just on the current state, but also on historical states [18]. As $\alpha \to 1^-$ in (3), the operator reverts to the traditional first-order derivative, representing the local change rate of the function with respect to time.

#### 2.1.1 Non-Markovian Random Walk Interpretation

We elucidate fractional-order derivatives by linking them to one-dimensional heat diffusion and memory-decaying non-Markovian random walks [28]. Assuming a random walker moves along an axis with infinitesimal intervals of space $\Delta x > 0$ and time $\Delta \tau > 0$, the walker moves a distance of $\Delta x$ from the current point $x$ in either direction with equal probability and waits at each location for a random period of time, a positive integer multiple of $\Delta \tau$. This introduces randomness in the waiting times between steps. We aim to compute $u(x, t)$, the probability of the walker arriving at position $x$ at time $t$. The waiting time distribution, $\psi_\alpha(n)$, is given by a power-law function $d_\alpha n^{-(1+\alpha)}$ with $d_\alpha > 0$ chosen to ensure $\sum_{n=1}^\infty \psi_\alpha(n) = 1$. The law of total probability is expressed as:

$$u(x, t) = \sum_{n=1}^\infty \left[ \frac{1}{2} u(x - \Delta x, t - n\Delta\tau) + \frac{1}{2} u(x + \Delta x, t - n\Delta\tau) \right] \psi_\alpha(n).$$

Here, the terms within brackets denote the probability of arriving at $x$ from either neighboring points, $x - \Delta x$ or $x + \Delta x$, each with probability $1/2$. The sum over $n$ accounts for the possibility that the walker could have remained stationary for an extended period $n\Delta\tau$ with a waiting time probability $\psi_\alpha(n)$. After subtracting $\sum_{n=1}^\infty \psi_\alpha(n)u(x, t - n\Delta\tau)$ from both sides and rearranging, we obtain:

$$\sum_{n=1}^\infty \frac{u(x, t) - u(x, t - n\Delta\tau)}{(n\Delta\tau)^{1+\alpha}}(\Delta\tau) = \frac{(\Delta x)^2}{2d_\alpha(\Delta\tau)^\alpha} \sum_{n=1}^\infty \delta_2 u(x, t - n\Delta\tau)\psi_\alpha(n). \tag{4}$$

where the second-order incremental quotient is defined as:

$$\delta_2 u(x,t) = \frac{u(x-\Delta x, t) + u(x + \Delta x, t) - 2u(x,t)}{(\Delta x)^2}.$$

In the limit as $\Delta x, \Delta \tau \to 0$ and assuming that $\frac{(\Delta x)^2}{d_\alpha (\Delta \tau)^\alpha} \to k_\alpha |\Gamma(-\alpha)|$ for a positive $k_\alpha$ [28], we obtain the time-fractional diffusion equation:

$$_\mathrm{M} D^\alpha u = \frac{k_\alpha}{2} u_{xx}, \tag{5}$$

where the summations on the left-hand side of (4) converge to the integration (2). As $\alpha \to 1^-$, (5) reverts to the standard heat diffusion equation:

$$\frac{\partial u(x,t)}{\partial t} = \frac{k_1}{2} u_{xx}. \tag{6}$$

Consequently, the aforementioned non-Markovian random walk with fading memory simplifies to the Markovian random walk, thereby eliminating the memory effects.

## 2.2  Integer-Order Continuous GNN Models

We denote an undirected graph as $\mathcal{G} = (\mathcal{V}, \mathbf{W})$, where $\mathcal{V}$ is the set of $|\mathcal{V}| = N$ nodes and $\mathbf{X} = \left([\mathbf{x}_1]^\mathsf{T}, \cdots, [\mathbf{x}_N]^\mathsf{T}\right)^\mathsf{T} \in \mathbb{R}^{N \times d}$ consists of rows $\mathbf{x}_i \in \mathbb{R}^{1 \times d}$ as node feature vectors. The $N \times N$ adjacency matrix $\mathbf{W} := (W_{ij})$ has elements $W_{ij}$ indicating the edge weight between the $i$-th and $j$-th nodes with $W_{ij} = W_{ji}$. In the subsequent GNNs inspired by dynamic processes, we let $\mathbf{X}(t) = \left([\mathbf{x}_1(t)]^\mathsf{T}, \ldots, [\mathbf{x}_N(t)]^\mathsf{T}\right)^\mathsf{T} \in \mathbb{R}^{N \times d}$ be the features at time $t$ with $\mathbf{X}(0) = \mathbf{X}$ serving as the initial condition. The time $t$ here acts as an analog to the layer index [7,30]. Typically, these dynamics can be described by:

$$\frac{\mathrm{d}\mathbf{X}(t)}{\mathrm{d}t} = \mathcal{F}(\mathbf{W}, \mathbf{X}(t)). \tag{7}$$

The function $\mathcal{F}$ is specifically tailored for graph dynamics as illustrated in Appendix F. For instance, in the GRAND model, $\mathcal{F}$ is defined as follows:

**GRAND** [7]: Drawing from the standard heat diffusion equation, GRAND formulates the following feature updating dynamics:

$$\frac{\mathrm{d}\mathbf{X}(t)}{\mathrm{d}t} = (\mathbf{A}(\mathbf{X}(t)) - \mathbf{I})\mathbf{X}(t), \tag{8}$$

where $\mathbf{A}(\mathbf{X}(t))$ is a learnable attention or fixed normalized matrix, and $\mathbf{I}$ is an identity matrix.

## 2.3  Fractional-Order Continuous GNN Models

Recently, the paper [15] introduces FROND, extending traditional integer-order graph neural differential equations such as (8), (40) and (42) to fractional-order equations. The framework is formalized as

$$D^\alpha \mathbf{X}(t) = \mathcal{F}(\mathbf{W}, \mathbf{X}(t)), \quad \alpha > 0, \tag{9}$$

where $\mathcal{F}$ represents the graph dynamics. Further, the study in [16] explores the robustness of FROND, demonstrating its ability to enhance the resilience of integer-order continuous GNNs under perturbations. This underscores the potential applications of FROND in various domains.

## 2.4  Motivation: Advanced Dynamics Modeling Capability

To intuitively understand the versatility and efficacy of the DRAGON framework in learning dynamics, we consider three classical stress-strain constitutive models for viscoelastic solids: the single-order Maxwell model [31], the multi-order Zener model [32], and the distributed-order Kelvin-Voigt model [33]. Using the FROND and DRAGON frameworks, we develop Neural Network(NN) methods to predict future states based on current observations.

Table 1: Comparison of MSE for the Maxwell, Zener, and Kelvin-Voigt models using FROND-NN and DRAGON-NN frameworks.

| Model | FROND-NN | DRAGON-NN |
|---|---|---|
| **Maxwell** [31] | $2.0 \times 10^{-4}$ | $5.6 \times 10^{-5}$ |
| **Zener** [32] | $3.6 \times 10^{-2}$ | $3.5 \times 10^{-3}$ |
| **Kelvin-Voigt** [33] | $3.3 \times 10^{-3}$ | $1.4 \times 10^{-4}$ |

The detailed descriptions and implementation specifics can be found in Appendix G.1. The results presented in Table 1 demonstrate that the DRAGON framework excels in fitting not only the multi-order model but also in capturing the dynamics of single-order and distributed-order models. We observe that the DRAGON framework achieves a Mean Squared Error (MSE) that is ten times smaller than that of the FROND method across all three models. This highlights the DRAGON framework's exceptional ability to effectively learn and adapt to a diverse range of dynamics, surpassing the capabilities of FROND.

## 3 DRAGON Framework

In this section, we introduce our general DRAGON framework for GNNs, with a random walk interpretation that elucidates the underlying mechanics when a specific diffusion-inspired system is utilized. Subsequently, we discuss numerical techniques for solving DRAGON. The versatility of our framework is highlighted by its capacity to encapsulate a broad spectrum of existing continuous GNN architectures, while simultaneously nurturing the development of more flexible continuous GNN designs within the research community in the future.

### 3.1 Framework

DRAGON generalizes the current integer-order and fractional-order continuous GNNs as it uses a learnable probability distribution over a range of real numbers for the fractional derivative orders. Consider a graph $\mathcal{G} = (\mathcal{V}, \mathbf{W})$ composed of $|\mathcal{V}| = N$ nodes with $\mathbf{W}$ being adjacency matrix as defined in Section 2.2. Similar to the approach used in integer-order continuous GNN models [5, 15] as presented in Section 2.2, we apply a preliminary learnable encoder function $\varphi : \mathcal{V} \rightarrow \mathbb{R}^d$ that maps each node to a feature vector. After stacking all these feature vectors, we obtain $\mathbf{X} \in \mathbb{R}^{N \times d}$. Employing the distributed-order fractional derivative outlined in (1), the feature dynamics in DRAGON are characterized by the following graph dynamic equation:

$$\int_a^b D^\alpha \mathbf{X}(t) \, \mathrm{d}\mu(\alpha) = \mathcal{F}(\mathbf{W}, \mathbf{X}(t)),  \tag{10}$$

where $[a, b]$ denotes the range of the order $\alpha$, $\mu$ is a learnable measure of $\alpha$, and $\mathcal{F}$ is a dynamic operator on the graph as illustrated in Appendix F.

**Remark 2.** *In practical engineering settings, the* Caputo *fractional derivative, represented by* $_\mathrm{C}D^\alpha$, *is commonly used [15, 17]. When leveraging the Caputo definition for the fractional derivative, as detailed in Section 3.3, the initial condition for (10) is given by* $\mathbf{X}^{[n]}(0) = \mathbf{X}$, *where* $\mathbf{X}^{[n]}(0)$ *denotes the* $n$-*th order derivative at* $t = 0$, *encompassing the initial node features for all integers* $n \in \mathbb{N} \cap [0, \lceil b \rceil]$ *[23]. Here,* $\lceil \cdot \rceil$ *is the ceiling function, and this setup ensures a unique solution [23]. For instance, when* $[a, b] = [0, 1]$, *we define the initial condition as* $\mathbf{X}(0) = \mathbf{X}$.

This framework generalizes prior continuous GNN models, encompassing them as special instances. Specifically, with $\mu(\alpha) = \delta(\alpha - 1)$, where $\delta$ is the Dirac delta function, (10) simplifies to a local first-order differential equation like [7, 8, 10–13]. When $[a, b] = [0, 2]$, we may obtain a distributed-order fractional wave propagation GNN model [21], which generalizes the second-order GraphCON model (40). When $\mu(\alpha) = \delta(\alpha - \alpha_o)$ for $\alpha_o \in \mathbb{R}^+$, (10) reduces to the FROND framework (9). Additionally, when $\mu$ adopts a discrete distribution over multiple integers, the model corresponds to TDE-GNN [14].

Following previous works, we set an integration time parameter $T$ to obtain $\mathbf{X}(T)$. The final node embeddings, employed for subsequent downstream tasks, can be decoded as $\zeta(\mathbf{X}(T))$, where $\zeta$ symbolizes a learnable decoder function.

### 3.2 Non-Markovian Graph Random Walk with Flexible Memory

In this subsection, we provide a non-Markovian graph random walk interpretation for DRAGON under a specific anomalous diffusion setting, where the dynamic operator $\mathcal{F}(\mathbf{W}, \mathbf{X}(t))$ in (10) is set as $(\mathbf{A}(\mathbf{X}(t)) - \mathbf{I})\mathbf{X}(t)$ in (8) with a fixed constant matrix $\mathbf{A}$. More specifically, we obtain the following linear distributed-order FDE:

$$\int_0^1 {}_\mathrm{M}D^\alpha \mathbf{X}(t) \, \mathrm{d}\mu(\alpha) = \mathbf{L}\mathbf{X}(t),  \tag{11}$$

where we set $\mathbf{A} = \mathbf{WD}^{-1}$ and $\mathbf{L} := \mathbf{WD}^{-1} - \mathbf{I}$ is the random walk Laplacian. Here, $\mathbf{D}$ is a diagonal matrix with $D_{ii} = d_i$, the degree of node $i$. For clarity, without loss of generality, similar to the approach in Section 2.1.1, we interpret $\mathbf{X}(t)$ as a $N$-dimensional probability or concentration vector $\mathbb{P}(t)$ over the graph nodes $\mathcal{V}$ at time $t$. The Marchaud–Weyl $_\mathrm{M}D^\alpha$ employed in (11) helps expedite the exposition of the subsequent random walk, drawing an analogy from the one-dimensional random walk discussed in Section 2.1.1.

For every individual value $\alpha_o \in (0, 1)$, we consider a random walker navigating over graph $\mathcal{G}$ with an infinitesimal interval of time $\Delta\tau > 0$. We assume that there is no self-loop in the graph topology. The dynamics of the random walk are characterized as follows:

1. The walker is expected to wait at the current location for a random period of time. The distribution of waiting times, $\psi_{\alpha_o}(n)$, is given by a power-law function $d_{\alpha_o} n^{-(1+\alpha_o)}$ with $d_{\alpha_o} > 0$ chosen to ensure $\sum_{n=1}^\infty \psi_{\alpha_o}(n) = 1$.
2. Upon deciding to make a jump, the walker can either move from the current node $i$ to a neighboring node $j$ with a probability of $(\Delta\tau)^{\alpha_o} d_{\alpha_o} |\Gamma(-\alpha_o)| \frac{W_{ij}}{d_i}$ if $i \neq j$. Alternatively, with a probability of $1 - (\Delta\tau)^{\alpha_o} d_{\alpha_o} |\Gamma(-\alpha_o)|$, it will remain at the current node $i$.

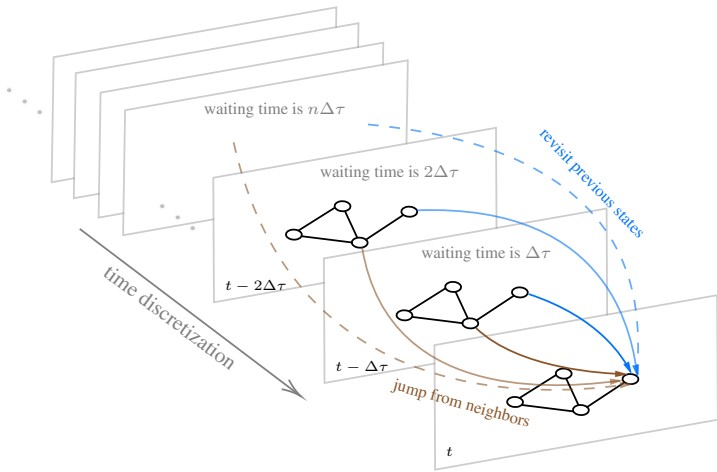

Figure 1: Visualization of the Non-Markovian Graph Random Walk. The diagram illustrates the walker's decision-making process during the walk. After waiting for a random duration $n\Delta$, the walker may either remain on the current node or proceed to a neighborhood node. This reflects the flexible, memory-influenced dynamics of the walker's movement.

We denote $\mathbb{P}_j(t; \alpha_o)$, the probability of the walker being at node $j$ at time $t$ with a specific order $\alpha_o$ and $\mu(\alpha) = \delta(\alpha - \alpha_o)$. The law of total probability is expressed as:

$$\mathbb{P}_j(t; \alpha_o) = \sum_{n=1}^\infty \left[ \sum_{\substack{i \in \mathcal{V} \\ i \neq j}} \mathbb{P}_i(t - n\Delta\tau; \alpha_o)(\Delta\tau)^{\alpha_o} d_{\alpha_o} |\Gamma(-\alpha_o)| \frac{W_{ij}}{d_i} \right. \tag{12}$$
$$\left. + \mathbb{P}_j(t - n\Delta\tau; \alpha_o)\big(1 - (\Delta\tau)^{\alpha_o} d_{\alpha_o} |\Gamma(-\alpha_o)|\big) \right] \psi_{\alpha_o}(n).$$

In this equation, the summation over $n$ accounts for the possibility that the walker may have remained stationary for a period of $n\Delta\tau$, with a waiting time probability of $\psi_{\alpha_o}(n)$. Fig. 1 provides a visualization of the non-Markovian graph random walk. For more explanation of the non-Markovian random walker on graphs, please refer to Appendix E. From (12), we can derive Theorem 1.

**Theorem 1.** *Given $\mu(\alpha) = \delta(\alpha - \alpha_o)$ where $\alpha_o \in (0, 1)$ and $\Delta\tau \to 0$, we establish that $\mathbb{P}(t; \alpha_o)$, the probability vector whose $j$-th element is $\mathbb{P}_j(t; \alpha_o)$, solves (11). That is to say, we have*

$$\int_0^1 {}_\mathrm{M}D^\alpha \mathbb{P}(t; \alpha_o) \, \mathrm{d}\mu(\alpha) = \mathbf{L}\mathbb{P}(t; \alpha_o). \tag{13}$$

**Remark 3.** *In Theorem 1, we present the graph random walk interpretation for the fractional anomalous diffusion equation (11) under the condition that $\mu = \delta(\alpha - \alpha_o)$. This condition represents a single-term fractional scenario similar to FROND. At its core, this type of random walk is non-Markovian, underscoring the importance of the entire walk history.*

From the discussion above, for a specific $\alpha_o$, the waiting time is steered by the power-law distribution $\propto n^{-(\alpha_o+1)}$. Moreover, the distributed-order fractional operator can be interpreted as a flexible superposition of the dynamics behaviors embodied by individual fractional-order operators. This generalization reframes the interpretation of graph random walk and enables more nuanced dynamics that accommodate diverse waiting times. Although it is feasible to formulate a random walk interpretation where the waiting time is linked to the measure $\mu$ and converges to the solution of (11), this approach relies on the intricate stopping time technique [34][Sec 7.5] and may sacrifice flexibility in waiting time insights. Instead, we propose a more modest conclusion, demonstrating that a weighted sum of $\psi_{\alpha_i}(n)$ can *approximate any waiting time*, highlighting the capability of our framework in comparison to FROND.

**Theorem 2.** *Let $C_0(\mathbb{N})$ be the space of functions on the natural numbers $\mathbb{N}$ vanishing at $\infty$, i.e., $f \in C_0(\mathbb{N})$ if and only $\lim_{n\to\infty} f(n) = 0$. Assume the sequence $(\alpha_m)_{m=1}^{\infty}$ is strict increasing in $[0, 1]$, then the span of $\{\psi_{\alpha_m}, m \geq 1\}$ is dense in $C_0(\mathbb{N})$ in the sense of uniform convergence.*

**Remark 4.** *Theorem 2 demonstrates the DRAGON framework's ability to approximate any waiting time distribution for graph random walkers, offering flexibility in modeling feature updating dynamics with varying extents of memory incorporation. This highlights the advantage of using DRAGON for deploying learnable and flexible feature updating dynamics. In contrast, FROND is confined to a fixed waiting time distribution, limiting its adaptability in modeling feature updating over time.*

### 3.3 Solving DRAGON

Previous continuous GNNs have leveraged neural ODE solvers [30] when $\mu = \delta(\alpha - 1)$. For example, in the explicit Euler scheme, neural ODEs are effectively reduced to residual networks with shared hidden layers [30]. Addressing the challenge of solving the distributed-order FDE (10) given by DRAGON, the standard approach involves discretizing it into a multi-term FDE. This is achieved by using a quadrature formula to approximate the integral term [21, 23]. As articulated in Sections 2.1 and 3.1, we follow the convention in the fractional calculus literature for real-world applications and employ the Caputo definition $_{\mathrm{C}}D^{\alpha}$ in this section. This choice is intuitive, as it seamlessly incorporates initial conditions into the problem as previously discussed under (10). The initial step is to approximate (10) as follows:

$$\sum_{j=0}^{n} w_{j\,\mathrm{C}}D^{\alpha_j}\mathbf{X}(t) = \mathcal{F}(\mathbf{W}, \mathbf{X}(t)) \tag{14}$$

where $\alpha_j \in [a, b]$, $j = 0, 1, \ldots, n$, are distinct interpolation points and $w_j$ are weights associated with the measure $\mu$. Reflecting the learnable nature of $\mu$, $w_j$ is directly set to be a learnable parameter in our implementation.

The next step is to solve the multi-term FDE presented in (14). According to the approach outlined in [17, Theorem 8.1], the multi-term FDE can be transformed into a system of single-order equations $_{\mathrm{C}}D^{\gamma}$, where $\gamma := 1/M$ and $M$ is the least common multiple of the denominators of $\alpha_0, \alpha_1, \ldots, \alpha_n$ when these coefficients are rational numbers. The classical fractional Adams–Bashforth–Moulton method can then be applied to solve the resulting system of single-order equations [15, 35]. This method is a generalization of the Euler scheme for ODEs to fractional scenarios (see Appendix C.3 for a detailed explanation).

An alternative approach involves directly approximating the fractional derivative operators as demonstrated in [36]. This discretization method can then be used to derive iterative methods for solving the multi-term FDE given in (14). Detailed procedures for this method are provided in Appendix C.4. Additionally, the approximation error analysis of the numerical solvers is discussed in Appendix D.

### 3.4 DRAGON GNNs

In Section 2.2 and Appendix F, several continuous GNNs, such as (8), (40) and (42), which employ integer-order derivatives, are introduced. We now extend these dynamical systems to operate under

our proposed DRAGON framework, which generalizes the scenarios to involve distributed-order fractional derivatives. More specifically, we present the following GNNs, which will be utilized in Section 4 to show the advantages of our framework over various graph benchmarks.

1. **D-GRAND:** By extending (8), we get

$$\int_0^1 D^\alpha \mathbf{X}(t) \, d\mu(\alpha) = (\mathbf{A}(\mathbf{X}(t)) - \mathbf{I})\mathbf{X}(t). \tag{15}$$

2. **D-GraphCON:** By extending (40), we get

$$\int_0^2 D^\alpha \mathbf{X}(t) \, d\mu(\alpha) = \sigma(\mathbf{F}_\theta(\mathbf{X}(t), t)) - \gamma \mathbf{X}(t). \tag{16}$$

3. **D-CDE:** By extending (42), we get

$$\int_0^1 D^\alpha \mathbf{X}(t) \, d\mu(\alpha) = (\mathbf{A}(\mathbf{X}(t)) - \mathbf{I})\mathbf{X}(t) + \mathrm{div}(\mathbf{V}(t) \circ \mathbf{X}(t)), \tag{17}$$

where $\mathrm{div}(\mathbf{V}(t) \circ \mathbf{X}(t))$ is given in (43) and (44).

Depending on the method used to compute the matrix $\mathbf{A}$ in (15), the D-GRAND model can be categorized into two versions: linear (D-GRAND-l) and non-linear (D-GRAND-nl). Similarly, based on the computation of $\mathbf{F}_\theta$ in (16), the D-GraphCON model also has two versions: linear (D-GraphCON-l) and non-linear (D-GraphCON-nl). Detailed explanations are provided in Appendix F.1.

## 4  Experiments

Our approach aims to enhance the capabilities of continuous GNN models by flexibly combining graph dynamics across different derivative orders. To achieve this, we have integrated DRAGON into several existing continuous GNN models and assessed their performance. Specifically, we conduct experiments on our proposed D-GRAND (15), D-GraphCON (16), and D-CDE (17) in this section, as well as D-GREAD and D-GRAND++ in Appendix I.3 and Appendix I.4.

### 4.1  Implementation Details

In our approach, we employ a fully connected (FC) layer as the encoder, $\varphi : \mathcal{V} \to \mathbb{R}^d$, to determine the initial values for DRAGON. Subsequently, another FC layer $\zeta$ serves as the decoder, transforming the output of DRAGON for downstream tasks. Most existing continuous GNNs are first-order or can be transformed into first-order representations of certain dynamic processes across graphs [9]. The FROND framework also restricts the fractional order to the range $[0, 1]$, maintaining identical initial conditions to those utilized in the original models. Given these considerations, we mainly restrict $\alpha_j$ values between $[0,1]$ in our implementation, while also balancing computational costs. The parameter $\alpha_j$ is selected to evenly divide the entire range, aiming to comprehensively cover values between $[0, 1]$. Typically, we set the number of $\alpha_j$ in (14) to 10. We also explore the empirical results when $\alpha_j$ exceeds 1, as shown in Appendix I.5. For a sensitivity analysis of the number

Table 2: Numerical results for various methods on LRGB tests.

| Method | Peptides-func Test AP ↑ | Peptides-Struct Test MAE ↓ |
|---|---|---|
| GCN [37] | 0.5930±0.0023 | 0.3496±0.0013 |
| GCNII [38] | 0.5543±0.0078 | 0.3471±0.0010 |
| GINE [39] | 0.5498±0.0079 | 0.3447±0.0045 |
| GatedGCN [40] | 0.5864±0.0077 | 0.3420±0.0013 |
| Transformer+LapPE [41] | 0.6326±0.0126 | 0.2529±0.0016 |
| SAN+LapPE [42] | 0.6384±0.0121 | 0.2683±0.0043 |
| SAN+RWSE [43] | 0.6439±0.0075 | 0.2545±0.0012 |
| GCN+DRew [44] | 0.6996±0.0076 | 0.2781±0.0028 |
| PathNN [45] | 0.6816±0.0026 | 0.2545±0.0032 |
| DRGNN [46] | 0.6586±0.0042 | 0.2495±0.0015 |
| GRAND-l | 0.6962±0.0015 | 0.2867±0.0009 |
| F-GRAND-l | 0.7126±0.0024 | 0.2677±0.0014 |
| D-GRAND-l | **0.7571±0.0014** | **0.2461±0.0014** |

and value of $\alpha_j$, we refer the readers to Appendix I.6. Details on the datasets used can be found in Appendix G.2.

### 4.2  Long Range Graph Benchmark

As illustrated in Remark 4, the DRAGON framework exhibits a distinctive intrinsic property: its ability to capture flexible memory effects, which is crucial for modeling long-range dependencies

in graph data [44]. To empirically validate this capability, we conduct experiments using the Long-Range Graph Benchmark (LRGB) [47]. Specifically, we focus on the Peptides molecular graphs dataset, performing *graph classification* on the Peptides-func dataset and *graph regression* based on the 3D structure of peptides in the Peptides-struct dataset. The performance metrics used are Average Precision (AP) for classification and Mean Absolute Error (MAE) for regression tasks. From Table 2, it is evident that the DRAGON framework outperforms the other methods on these two long-range graph datasets, even when compared to state-of-the-art (SOTA) techniques. Notably, DRAGON achieves an improvement of approximately 4~6% over traditional continuous GNNs like GRAND-l and F-GRAND-l. This demonstrates DRAGON's capability to effectively capture long-range dependencies in graph data.

## 4.3 Node Classification

### 4.3.1 Homophilic Graph Datasets

In our evaluation on homophilic datasets, we leverage a diverse set of datasets including citation networks (Cora [48], Citeseer [49], Pubmed [50]), tree-structured datasets (Disease and Airport [51]), as well as coauthor and co-purchasing graphs (CoauthorCS [52], Computer and Photo [53]). For the Disease and Airport datasets, we follow the data partitioning and preprocessing procedures as described in [51]. For all other datasets, we adopt random splits for the largest connected component (LCC), in line with the approach detailed in [7].

Table 3: Node classification results(%) for random train-val-test splits. The best result of **each** continuous GNN family is highlighted in **red**.

| Method | Cora | Citeseer | Pubmed | CoauthorCS | Computer | Photo | CoauthorPhy | Airport | Disease |
|---|---|---|---|---|---|---|---|---|---|
| GCN [37] | 81.5±1.3 | 71.9±1.9 | 77.8±2.9 | 91.1±0.5 | 82.6±2.4 | 91.2±1.2 | 92.8±1.0 | 81.6±0.6 | 69.8±0.5 |
| GAT [54] | 81.8±1.3 | 71.4±1.9 | 78.7±2.3 | 90.5±0.6 | 78.0±19.0 | 85.7±20.3 | 92.5±0.9 | 81.6±0.4 | 70.4±0.5 |
| HGCN [51] | 78.7±1.0 | 65.8±2.0 | 76.4±0.8 | 90.6±0.3 | 80.6±1.8 | 88.2±1.4 | 90.8±1.5 | 85.4±0.7 | 89.9±1.1 |
| GIL [55] | 82.1±1.1 | 71.1±1.2 | 77.8±0.6 | 89.4±1.5 | – | 89.6±1.3 | – | 91.5±1.7 | 90.8±0.5 |
| GRAND-l | 83.6±1.0 | 73.4±0.5 | 78.8±1.7 | 92.9±0.4 | 83.7±1.2 | 92.3±0.9 | 93.5±0.9 | 80.5±9.6 | 74.5±3.4 |
| F-GRAND-l | 84.8±1.1 | 74.0±1.5 | 79.4±1.5 | 93.0±0.3 | 84.4±1.5 | 92.8±0.6 | 94.5±0.4 | 98.1±0.2 | 92.4±3.9 |
| D-GRAND-l | **85.1±1.3** | **74.5±1.1** | **79.6±2.6** | **93.2±0.3** | **87.3±1.3** | **93.1±0.8** | **94.6±0.2** | **98.5±0.1** | **93.2±2.5** |
| GRAND-nl | 82.3±1.6 | 70.9±1.0 | 77.5±1.8 | 92.4±0.3 | 82.4±2.1 | 92.4±0.8 | 91.4±1.3 | 90.9±1.6 | 81.0±6.7 |
| F-GRAND-nl | 83.2±1.1 | 74.7±1.9 | 79.2±0.7 | 92.9±0.4 | 84.1±0.9 | 93.1±0.9 | 93.9±0.5 | 96.1±0.7 | 85.5±2.5 |
| D-GRAND-nl | **83.9±1.3** | **74.8±1.6** | **79.5±2.6** | **93.1±0.3** | **87.1±1.0** | **93.4±0.5** | **94.3±0.6** | **97.7±0.4** | **89.3±2.7** |
| GraphCON-l | 81.9±1.7 | 72.9±2.1 | 78.8±2.6 | 92.3±0.3 | 84.9±0.5 | 90.8±1.8 | 93.9±0.4 | 68.6±2.1 | 87.5±4.1 |
| F-GraphCON-l | 84.6±1.4 | **75.3±1.1** | 80.3±1.3 | 92.8±0.4 | 86.2±0.8 | 93.3±1.0 | 94.1±0.5 | 97.3±0.5 | 92.1±2.8 |
| D-GraphCON-l | **84.6±1.3** | 74.4±1.4 | **80.7±1.6** | **92.9±0.3** | **86.9±1.0** | **93.7±0.4** | 94.3±0.5 | **98.3±0.2** | **93.3±2.1** |
| GraphCON-nl | 83.2±1.4 | 73.2±1.8 | 79.5±1.8 | 88.7±0.9 | 79.2±1.1 | 85.5±2.3 | 93.1±0.3 | 74.1±2.7 | 65.7±5.9 |
| F-GraphCON-nl | 83.9±1.2 | 73.4±1.5 | 79.4±1.3 | 90.4±0.6 | 83.6±2.2 | **94.1±0.7** | 93.0±0.6 | 97.3±0.8 | 86.9±4.0 |
| D-GraphCON-nl | **84.2±1.2** | **74.0±2.1** | **79.5±1.1** | **92.0±0.2** | **87.1±1.0** | 93.8±0.8 | **94.0±0.4** | **98.3±0.3** | **91.4±1.6** |

Table 4: Node classification results(%). The best and the second-best result for each criterion are highlighted in **red** and **blue**, respectively.

| Method $h_{\text{adj}}$ | Roman-empire -0.05 | Wiki-cooc -0.03 | Minesweeper 0.01 | Questions 0.02 | Workers 0.09 | Amazon-ratings 0.14 |
|---|---|---|---|---|---|---|
| ResNet [56] | 65.71±0.44 | 89.36±0.71 | 50.95±1.12 | 70.10±0.75 | 73.08±1.28 | 45.70±0.69 |
| H2GCN [57] | 68.09±0.29 | 89.24±0.32 | 89.95±0.38 | 66.66±1.84 | 81.76±0.68 | 41.36±0.47 |
| CPGNN [58] | 63.78±0.50 | 84.84±0.66 | 71.27±1.14 | 67.09±2.63 | 72.44±0.80 | 44.36±0.35 |
| GPR-GNN [59] | 73.37±0.68 | 91.90±0.78 | 81.79±0.99 | 73.41±1.24 | 70.59±1.15 | 43.90±0.48 |
| GloGNN [60] | 63.85±0.49 | 88.49±0.45 | 62.53±1.34 | 67.15±1.92 | 73.90±0.95 | 37.28±0.66 |
| FAGCN [61] | 70.53±0.99 | 91.88±0.37 | 89.69±0.60 | **77.04±1.56** | 81.87±0.94 | 46.32±2.50 |
| GBK-GNN [62] | 75.87±0.43 | 97.81±0.32 | 83.56±0.84 | 72.98±1.05 | 78.06±0.91 | 43.47±0.51 |
| ACM-GCN [63] | 68.35±1.95 | 87.48±1.06 | 90.47±0.57 | OOM | 78.25±0.78 | 38.51±3.38 |
| GRAND [7] | 71.60±0.58 | 92.03±0.46 | 76.67±0.98 | 70.67±1.28 | 75.33±0.84 | 45.05±0.65 |
| GraphBel [10] | 69.47±0.37 | 90.30±0.50 | 76.51±1.03 | 70.79±0.99 | 73.02±0.92 | 43.63±0.42 |
| Diag-NSD [64] | 77.50±0.67 | 92.06±0.40 | 89.59±0.61 | 69.25±1.15 | 79.81±0.99 | 37.96±0.20 |
| ACMP [65] | 71.27±0.59 | 92.68±0.37 | 76.15±1.12 | 71.18±1.03 | 75.03±0.92 | 44.76±0.52 |
| TDE-GNN [14] | 64.29±0.58 | 84.95±0.78 | 61.15±2.24 | 68.94±1.69 | 75.13±0.81 | 40.33±1.37 |
| CDE [12] | 91.64±0.28 | 97.99±0.38 | 95.50±5.23 | 75.17±0.99 | 80.70±1.04 | 47.63±0.43 |
| F-CDE [15] | **93.06±0.55** | **98.73±0.68** | **96.04±0.25** | 75.17±0.99 | **82.68±0.86** | **49.01±0.56** |
| D-CDE | **93.87±0.41** | **98.58±0.12** | **96.47±1.89** | **75.53±0.98** | **83.02±0.86** | **49.43±1.26** |

### 4.3.2 Heterophilic Graph Datasets

For evaluating performance on heterophilic datasets, we utilize six datasets introduced in [66], with details provided in Appendix G.2. As highlighted in [66], these datasets are characterized by lower adjusted homophily $h_{\mathrm{adj}}$, indicating a higher degree of heterophily. In our experimental setup with these heterophilic datasets, we follow the data splitting strategy described in [66], dividing the data into 50% for training, 25% for validation, and 25% for testing.

### 4.3.3 Performance of DRAGON framework

As shown in Table 3, for homophilic datasets such as citation networks, coauthor networks, and co-purchasing networks, our DRAGON framework enhances the performance of continuous backbones like GRAND and GraphCON. This demonstrates the ability of our DRAGON framework to seamlessly integrate with existing continuous GNNs and improve their performance. Notably, on tree-structured datasets, our DRAGON framework significantly boosts the performance of both GRAND and GraphCON. In particular, on the Airport dataset, our DRAGON framework excels, achieving a 7% performance increase compared to the GIL model specifically designed for this type of tree-like dataset. Compared to FROND, our DRAGON framework shows improvements on most datasets. The results of the graph node classification on heterophilic datasets are presented in Table 4. As indicated in Table 4, the proposed D-CDE model with our DRAGON framework improves the performance of the original CDE and F-CDE models on five out of the six datasets. This underscores the ability of DRAGON to capture flexible memory effects as proved in Theorem 2, highlighting its enhanced capability in modeling complex feature updating dynamics.

## 4.4 Model Complexity

For the Adams-Bashforth-Moulton method (25), the numerical solution is computed iteratively for $E := T/h$ time steps, where $h$ represents the discretization size and $T$ the integration time. This process involves repeated computation of $\mathcal{F}(\mathbf{W}, \mathbf{X}_j)$ for each iteration. By storing intermediate function evaluation values $\{\mathcal{F}(\mathbf{W}, \mathbf{X}_j)\}_j$, we can express the total computational time complexity across the process as $\sum_{k=0}^{E}(C + O(k))$, where $O(k)$ indicates the computational overhead from summing and weighting the $k$ terms at each step. Here, $C$ represents the complexity of computing $\mathcal{F}$. This yields a total cost of $O\left(EC + E^2\right)$. If a fast algorithm for the convolution computations is available, we typically require $O(E \log E)$ for the convolution [67], resulting in $O(EC + E \log E)$. If the cost of weighted summing is minimal, the complexity is reduced to $O(EC)$. For the Grünwald-Letnikov method (32), the computational complexity is the same as that of the method (25).

The term $C$ denotes the computational complexity of the function $\mathcal{F}$. For instance, setting $\mathcal{F}$ to the GRAND model results in $C = |\mathcal{E}|d$, where $|\mathcal{E}|$ represents the edge set size and $d$ the dimensionality of the features [7]. Alternatively, using the GREAD model results in $C = O((|\mathcal{E}| + |\mathcal{E}_2|)d + |\mathcal{E}|d_{\max})$, where $|\mathcal{E}_2|$ accounts for the number of two-hop edges, and $d_{\max}$ is the maximum degree among nodes [11]. More details of the computation cost can be found in Appendix H.

## 5 Conclusion

We introduce the DRAGON framework, which incorporates distributed-order fractional derivatives into continuous GNNs. DRAGON advances the field by employing a learnable distribution of fractional derivative orders, surpassing the constraints of existing continuous GNN models. This approach eliminates the need for fine-tuning the fractional order, as required in FROND, and enriches the dynamics and representational capacity of existing continuous GNN models. We also provide a flexible random walk interpretation. Through rigorous empirical testing, DRAGON has demonstrated not only its adaptability but also its consistent outperformance compared to other continuous GNN models. Consequently, DRAGON establishes itself as a powerful framework for advancing graph-related tasks.

## Acknowledgments and Disclosure of Funding

This research is supported by the National Research Foundation, Singapore and Infocomm Media Development Authority under its Future Communications Research and Development Programme. Xuhao Li is supported by National Natural Science Foundation of China (Grant No. 12301491). The computational work for this article was partially performed on resources of the National Supercomputing Centre, Singapore (https://www.nscc.sg). To improve the readability, parts of this paper have been grammatically revised using ChatGPT [68].

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

## A    Introduction

This supplementary material complements the main body of our paper by providing additional details and supporting evidence for the assertions made therein. The structure of this document is organized as follows:

1. A comprehensive background on fractional calculus is detailed in Appendix B.

2. Details of the FDE solvers used in our paper are outlined in Appendix C, along with the corresponding approximation error analysis for the solvers in Appendix D.

3. Additional explanations for the non-Markovian random walk interpretation are provided in Appendix E.

4. An extended introduction to traditional integer-order continuous GNNs from the literature is presented in Appendix F.

5. Additional implementation details, dataset specifics, and model complexity are elaborated in Appendices G and H.

6. More experimental results are available in Appendix I.

7. Theoretical results from the main paper are rigorously proven in Appendix J.

8. Limitations and broader impacts are discussed in Appendix K.

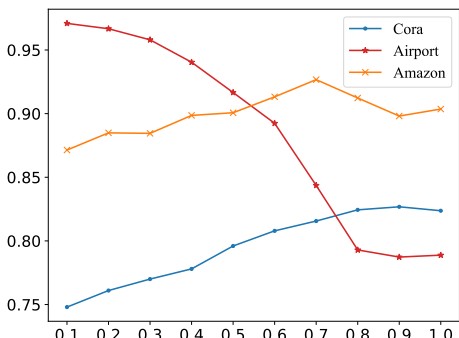

Figure 2: Variation of test accuracy with fractional order $\alpha$ in the FROND model

## B    Caputo Fractional Derivative

In our study, we introduce two definitions for fractional derivatives. While the elegance and interpretability of the Marchaud–Weyl derivative, especially its connection to random walks, is thoroughly discussed in the main paper, the practical realm of engineering often gravitates towards the Caputo fractional derivative, denoted as $_CD^\alpha$ [17]. Our alignment with the fractional calculus literature leads us to adopt the Caputo definition in Section 3.3. This preference stems from the inherent advantage of the Caputo derivative: it naturally integrates initial conditions, as elaborated in (10). The two definitions are equivalent under certain constraints [17, 29].

Below, we explore further the details of the Caputo fractional derivative to provide readers with a deeper understanding. For notational simplicity in this supplementary material, except in Appendix J, we use $D^\alpha$ interchangeably with $_CD^\alpha$, as we solely focus on the Caputo definition in this context.

The Caputo fractional derivative of a function $f(t)$ over an interval $[0, T]$, of a general positive order $\alpha \in (0, \infty)$, is defined as follows:

$$D^\alpha f(t) = \frac{1}{\Gamma(\lceil\alpha\rceil - \alpha)} \int_0^t (t-\tau)^{\lceil\alpha\rceil-\alpha-1} f^{[\lceil\alpha\rceil]}(\tau)\mathrm{d}\tau, \tag{18}$$

Here, $\lceil\alpha\rceil$ is the smallest integer greater than or equal to $\alpha$, $\Gamma(\cdot)$ symbolizes the gamma function, and $f^{[\lceil\alpha\rceil]}(\tau)$ signifies the $\lceil\alpha\rceil$-order derivative of $f$. Within this definition, it is presumed that

$f^{[\lceil \alpha \rceil]} \in L^1[0, T]$, i.e., $f^{[\lceil \alpha \rceil]}$ is Lebesgue integrable, to ensure the well-defined nature of $D^\alpha f(t)$ as per (18) [17]. When addressing a vector-valued function, the Caputo fractional derivative is defined on a component-by-component basis for each dimension, similar to the integer-order derivative. For ease of exposition, we explicitly handle the scalar case here, although all following results can be generalized to vector-valued functions. The Laplace transform for a general order $\alpha \in (0, \infty)$ is presented in [17, Theorem 7.1] as:

$$\mathcal{L}D^\alpha f(s) = s^\alpha \mathcal{L}f(s) - \sum_{k=1}^{\lceil \alpha \rceil} s^{\alpha-k} f^{[k-1]}(0). \tag{19}$$

where we assume that $\mathcal{L}f$ exists on $[s_0, \infty)$ for some $s_0 \in \mathbb{R}$. In contrast, for the integer-order derivative $f^{[\alpha]}$ when $\alpha$ is a positive integer, we also have the formulation (19), with the only difference being the range of $\alpha$. Therefore, as $\alpha$ approaches some integer, the Laplace transform of the Caputo fractional derivative converges to the Laplace transform of the traditional integer-order derivative. As a result, we can conclude that *the Caputo fractional derivative operator generalizes the traditional integer-order derivative* since their Laplace transforms coincide when $\alpha$ takes an integer value. Furthermore, the inverse Laplace transform indicates the uniquely determined $D^\alpha f = f^{[\alpha]}$ (in the sense of almost everywhere [69]).

Under specific reasonable conditions, we can directly present this generalization as follows. We suppose $f^{[\lceil \alpha \rceil]}(t)$ (18) is continuously differentiable. In this context, integration by parts can be utilized to demonstrate that

$$D^\alpha f(t) = \frac{1}{\Gamma(\lceil \alpha \rceil - \alpha)} \left( - \left[ f^{[\lceil \alpha \rceil]}(\tau) \frac{(t-\tau)^{\lceil \alpha \rceil - \alpha}}{\lceil \alpha \rceil - \alpha} \right] \Big|_0^t + \int_0^t f^{[\lceil \alpha \rceil + 1]}(\tau) \frac{(t-\tau)^{\lceil \alpha \rceil - \alpha}}{\lceil \alpha \rceil - \alpha} \mathrm{d}\tau \right)$$

$$= \frac{t^{\lceil \alpha \rceil - \alpha} f^{[\lceil \alpha \rceil]}(0)}{\Gamma(\lceil \alpha \rceil - \alpha + 1)} + \frac{1}{\Gamma(\lceil \alpha \rceil - \alpha + 1)} \times \int_0^t (t-\tau)^{\lceil \alpha \rceil - \alpha} f^{[\lceil \alpha \rceil + 1]}(\tau) \mathrm{d}\tau. \tag{20}$$

When $\alpha \to \lceil \alpha \rceil$, we get the following

$$\lim_{\alpha \to \lceil \alpha \rceil} D^\alpha f(t) = f^{[\lceil \alpha \rceil]}(0) + \int_0^t f^{[\lceil \alpha \rceil + 1]}(\tau) \mathrm{d}\tau$$

$$= f^{[\lceil \alpha \rceil]}(0) + f^{[\lceil \alpha \rceil]}(t) - f^{[\lceil \alpha \rceil]}(0) \tag{21}$$

$$= f^{[\lceil \alpha \rceil]}(t).$$

In parallel to the integer-order derivative, given *certain conditions* ( [17, Lemma 3.13]), the Caputo fractional derivative possesses the semigroup property:

$$D^\varepsilon D^n f = D^{n+\varepsilon} f. \tag{22}$$

Note, however, that in general, the Caputo fractional derivative does not possess semigroup property [17, Lemma 3.12]. The Caputo fractional derivative also exhibits linearity, but does not adhere to the same Leibniz and chain rules as its integer counterpart. As such properties are not utilized in our work, we refer interested readers to [17, Theorem 3.17 and Remark 3.5.]. We believe the above explanation facilitates understanding the relation between the Caputo derivative and its generalization of the integer-order derivative.

## C Numerical Solvers for FDEs

In this section, we introduce basic single-term FDEs along with techniques for solving them. We also discuss multi-term FDEs and describe methods to convert them into single-term FDEs. In our paper, we approximate the distributed-order FDE (10) using the multi-term FDE (14). We present two techniques to solve the multi-term FDE (14): one technique directly uses the single-term FDE solver, while the other approximates each fractional differential operator. For conditions necessary for the existence and uniqueness of solutions for single- and multi-term FDEs, we direct interested readers to [17, Chapter 6 and 8] and [15].

## C.1 Single-Term Solver

A single-term FDE is represented as:

$$D^\alpha y(t) = f(t, y(t)) \tag{23}$$

where the initial conditions take the form:

$$D^k y(0) = y_0^{[k]}, \quad k = 0, 1, \ldots, \lceil \alpha \rceil - 1. \tag{24}$$

with $y_0^{[k]}$ representing the $k$-order derivative at point $0$.

Our approach to solving (23) is based on the fractional Adams–Bashforth–Moulton method described in [70]. The basic predictor $y_{k+1}$ is expressed as:

$$y_{k+1} = \sum_{j=0}^{\lceil \alpha \rceil - 1} \frac{t_{k+1}^j}{j!} y_0^{[j]} + \frac{1}{\Gamma(\alpha)} \sum_{j=0}^{k} b_{j,k+1} f(t_j, y_j). \tag{25}$$

Here, $k$ denotes the current iteration or time step index in the discretization process, $h$ is the step size or time interval between successive approximations with $t_j = hj$, and $y_j$ is the numerical approximation of $y(t_j)$. $\lceil \cdot \rceil$ represents the ceiling function, and when $0 < \alpha \leq 1$, $\lceil \alpha \rceil = 1$. The coefficients $b_{j,k+1}$ are defined as follows:

$$b_{j,k+1} = \frac{h^\alpha}{\alpha} \left( (k + 1 - j)^\alpha - (k - j)^\alpha \right), \tag{26}$$

Using this predictor, it is possible to derive a corrector term to improve the accuracy of the solver. Nonetheless, we omit this corrector term in this work and leave its detailed exploration and implications for DRAGON to subsequent studies.

## C.2 Convert Multi-Term to Single-Term

We reference a theorem from [17] which provides a method to transform multi-term FDEs into their single-term counterparts, specifically when dealing with rational numbers.

**Theorem 3.** *[17, Theorem 8.1.] Consider the equation*

$$D_t^{n_k} y(x) = f\left(x, y(x), D_t^{n_1} y(x), D_t^{n_2} y(x), \ldots, D_t^{n_{k-1}} y(x)\right), \tag{27}$$

*subject to the initial conditions*

$$y^{[j]}(0) = y_0^{[j]}, \quad j = 0, 1, \ldots, \lceil n_k \rceil - 1,$$

*where $n_k > n_{k-1} > \ldots > n_1 > 0, n_j - n_{j-1} \leq 1$ for all $j = 2, 3, \ldots, k$ and $0 < n_1 \leq 1$. Assume that $n_j \in \mathbb{Q}$ for all $j = 1, 2, \ldots, k$, define $M$ to be the least common multiple of the denominators of $n_1, n_2, \ldots, n_k$ and set*

$$\gamma := 1/M \text{ and } N := Mn_k.$$

*Then this initial value problem is equivalent to the system of equations*

$$
\begin{aligned}
D_t^\gamma y_0(x) &= y_1(x), \\
D_t^\gamma y_1(x) &= y_2(x), \\
&\vdots \\
D_t^\gamma y_{N-2}(x) &= y_{N-1}(x), \\
D_t^\gamma y_{N-1}(x) &= f\left(x, y_0(x), y_{n_1/\gamma}(x), \ldots, y_{n_{k-1}/\gamma}(x)\right),
\end{aligned}
\tag{28}
$$

*together with the initial conditions*

$$y_j(0) = \begin{cases} y_0^{[j/M]}, & \text{if } j/M \in \mathbb{N}_0, \\ 0, & \text{else}, \end{cases}$$

*in the following sense:*

(a) *Whenever* $Y := (y_0, \ldots, y_{N-1})^{\mathsf{T}}$ *with* $y_0 \in C^{\lceil n_k \rceil}[0, T]$ *for some* $c > 0$ *is the solution of the system* (28), *the function* $y := y_0$ *solves the multi-term equation initial value problem* (27). *Here, the notation* $C^m[0, T]$ *denotes the space of functions that have a continuous* $m$*-th derivative.*

(b) *Whenever* $y \in C^{\lceil n_k \rceil}[0, T]$ *is a solution of the multi-term initial value problem* (27), *the vector function* $Y := (y_0, \ldots y_{N-1})^{\mathsf{T}} := \left( y, D_t^\gamma y, D_t^{2\gamma} y, \ldots, D_t^{(N-1)\gamma} y \right)^{\mathsf{T}}$ *solves the multidimensional initial value problem* (28).

## C.3 Solution Strategy I for (14)

Utilizing the theorem mentioned earlier from [17], we can address the solution of (14) as presented in the main manuscript. Specifically, we can express (14) as

$$w_n D^{\alpha_n} \mathbf{X}(t) = \mathcal{F}(\mathbf{W}, \mathbf{X}(t)) - \sum_{j=0}^{n-1} w_j D^{\alpha_j} \mathbf{X}(t). \tag{29}$$

Subsequently, the single-term solver (25) and Theorem 3 can be employed to solve this equation.

## C.4 Solution Strategy II for (14)

Consider the general multi-term (or more precisely, $n$-term) fractional differential equation:

$$\sum_{j=0}^{n} w_j D^{\alpha_j} y(t) = f(t, y(t)), \tag{30}$$

with initial condition $y(0) = y_0$, where $w_j$ are coefficients, $\alpha_j \in (0, 1)$ are fractional orders, and $f(t)$ is a given function.

Divide the interval $[0, T]$ into $E$ equally spaced points with step size $h$:

$$t_i = ih, \quad i = 0, 1, 2, \ldots, E,$$

where $h = \frac{T}{E}$. The Grünwald-Letnikov approximation for the fractional derivative $D^\alpha y(t)$ with $\alpha \in (0, 1)$ is given by:

$$D^\alpha y(t_i) \approx \frac{1}{h^\alpha} \sum_{k=0}^{i} (-1)^k \binom{\alpha}{k} [y(t_{i-k}) - y_0], \tag{31}$$

where $\binom{\alpha}{k}$ is the binomial coefficient for non-integer $\alpha$:

$$\binom{\alpha}{k} = \frac{\Gamma(\alpha + 1)}{\Gamma(k + 1)\Gamma(\alpha - k + 1)}.$$

The fractional derivative $D^{\alpha_j} y(t_i)$ for each $\alpha_j$ can be approximated as:

$$D^{\alpha_j} y(t_i) \approx \frac{1}{h^{\alpha_j}} \sum_{k=0}^{i} (-1)^k \binom{\alpha_j}{k} [y(t_{i-k}) - y_0].$$

We then combine the terms for the multi-term FDE:

$$\sum_{j=0}^{n} w_j \frac{1}{h^{\alpha_j}} \sum_{k=0}^{i} (-1)^k \binom{\alpha_j}{k} [y(t_{i-k}) - y_0] = f(t_{i-1}, y(t_{i-1}))$$

, or equivalently,

$$\sum_{j=0}^{n} w_j \frac{1}{h^{\alpha_j}} \sum_{k=1}^{i} (-1)^k \binom{\alpha_j}{k} [y(t_{i-k}) - y_0] + \sum_{j=0}^{n} w_j \frac{1}{h^{\alpha_j}} y(t_i) - \sum_{j=0}^{n} w_j \frac{1}{h^{\alpha_j}} y_0$$
$$= f(t_{i-1}, y(t_{i-1}))$$

Finally, denoting the approximation of $y(t_i)$ as $y_i$ at each iteration, for each $i$ from 1 to $E := T/h$, we update the numerical solution $y_i$ using:

$$y_i = \frac{f(t_{i-1}, y_{i-1}) + \sum_{j=0}^{n} w_j \frac{1}{h^{\alpha_j}} y_0 - \sum_{j=0}^{n} w_j \frac{1}{h^{\alpha_j}} \sum_{k=1}^{i} (-1)^k \binom{\alpha_j}{k} [y_{i-k} - y_0]}{\sum_{j=0}^{n} w_j \frac{1}{h^{\alpha_j}}} \tag{32}$$

This provides a step-by-step approach to iteratively update the solution of the $n$-term FDE using the Grünwald-Letnikov approximation for fractional derivatives.

Substituting the (32) into (14), we obtain the numerical solution:

$$\mathbf{X}_i = \frac{\mathcal{F}(\mathbf{W}, \mathbf{X}_{i-1}) + \sum_{j=0}^{n} w_j \frac{1}{h^{\alpha_j}} \mathbf{X}_0 - \sum_{j=0}^{n} w_j \frac{1}{h^{\alpha_j}} \sum_{k=1}^{i} (-1)^k \binom{\alpha_j}{k} [\mathbf{X}_{i-k} - \mathbf{X}_0]}{\sum_{j=0}^{n} w_j \frac{1}{h^{\alpha_j}}} \tag{33}$$

where $\mathbf{X}_i$ is numerical approximation of $\mathbf{X}(t_i)$.

## D  Approximation Error

As discussed in Section 3.3, solving the distributed-order FDE as specified in (10) involves two primary steps:

1. Discretizing the distributed-order derivative using a classical quadrature rule. For instance, assuming $w(\alpha) = \mu'(\alpha)$, the application of the composite Trapezoid rule [71, 72] yields:

$$\int_a^b D^\alpha \mathbf{X}(t) \, \mathrm{d}\mu(\alpha) = \frac{\Delta\alpha}{2} \left[ w(\alpha_0) D^{\alpha_0} \mathbf{X}(t) + 2 \sum_{j=1}^{n-1} w(\alpha_j) D^{\alpha_j} \mathbf{X}(t) + w(\alpha_n) D^{\alpha_n} \mathbf{X}(t) \right] + O((\Delta\alpha)^2), \tag{34}$$

where $\Delta\alpha = (b-a)/n$ and $\alpha_j = a + j\Delta\alpha$. After omitting smaller terms, this approximation leads to the multi-term FDE presented in (14).

2. Solving (14) using the fractional Adams–Bashforth–Moulton method as described in (25) or the Grünwald-Letnikov method as specified in (32).

Therefore, the approximation error of the true solution comprises the numerical quadrature error in Step 1 and the numerical solver error in Step 2. The quadrature error is directly evidenced by (34). To address the solver error, we consider the general $n$-term FDE as detailed in (30).

For the fractional Adams–Bashforth–Moulton method described in (25), the multi-term FDEs are transformed into a system of single-term equations. This system is then solved using the method specified in (25). The approximation error for this solver is quantified as follows [35]:

$$\max_{j=0,1,...,E} |y(t_j) - y_j| = O(h^{1+\min\{\alpha_j\}}), \tag{35}$$

where $y_j$ denotes the value of the solution at time $t_j$ as computed by the numerical method, and $y(t_j)$ represents the exact solution at time $t_j$, $h$ is the step size.

For the Grünwald-Letnikov method detailed in (32), we apply the Grünwald-Letnikov approximation [73] to each fractional derivative $D^{\alpha_j} y(t)$, which is computed as:

$$D^{\alpha_j} y(t_i) = \frac{1}{h^{\alpha_j}} \sum_{k=0}^{i} (-1)^k \binom{\alpha_j}{k} [y(t_{i-k}) - y_0] + O(h).$$

Utilizing correction techniques detailed in [74], the approximation error is calculated as:

$$\max_{j=0,1,...,E} |y(t_j) - y_j| = O(h), \tag{36}$$

Thus, the total error is a cumulative measure of the approximation errors from both Step 1 and Step 2.

# E   Non-Markovian Graph Random Walk Interpretation

Section 3.2 details the dynamics of the random walk. For enhanced clarity, here we include the corresponding transition probability representation for the non-Markovian random walker at time $t$, which explicitly accounts for node positions throughout the entire path history $(\ldots, q(t - n\Delta\tau), \ldots, q(t - \Delta\tau))$. Here, $q(t)$ represents the walker's position on the graph nodes $\{1, 2, \ldots, |\mathcal{V}|\}$ at time $t$. This model ensures that all historical states influence transitions, emphasizing the model's non-Markovian nature. We consider a random walker navigating over graph $\mathcal{G}$ with an infinitesimal interval of time $\Delta\tau > 0$. We assume that there is no self-loop in the graph topology.

For every individual value $\alpha_o \in (0, 1)$, the transition probability of the random walk dynamics as described above Fig. 1 is characterized as follows:

$$\mathbb{P}\left(q(t) = j_t \mid \ldots, q(t - n\Delta\tau) = j_{t-n\Delta\tau}, \ldots, q(t - \Delta\tau) = j_{t-\Delta\tau}\right)$$

$$= \begin{cases} (1 - K)\,\psi_{\alpha_o}(n) & \text{if revisiting historical positions } q(t - n\Delta\tau) \text{ with } j_t = j_{t-n\Delta\tau}, \text{ i.e., the} \\ & \text{walker's wait time is } n\Delta\tau \text{ and stays at the same node,} \\[2ex] \left(K\dfrac{W_{j_{t-\Delta\tau}j_t}}{d_{j_{t-\Delta\tau}}}\right)\psi_{\alpha_o}(n) & \text{if jumping from historical positions } j_{t-n\Delta\tau} \text{ to } j_t, \text{ i.e., the walker's wait.} \\ & \text{time is } n\Delta\tau \text{ and jumps to } j_{t-n\Delta\tau}\text{'s neighbour } j_t \end{cases}$$

$$\tag{37}$$

where $K := (\Delta\tau)^{\alpha_o} d_{\alpha_o} |\Gamma(-\alpha_o)|$ is a normalization coefficient, $j_{t-n\Delta\tau}$ is the node index visited at time $t - n\Delta\tau$, and $\psi_{\alpha_o}(n)$ is the probability that the walker's waiting time is $n\Delta\tau$. For a specific $\alpha_o$, the waiting time $\psi_{\alpha_o}(n)$ follows a power-law distribution $\propto n^{-(\alpha_o+1)}$. Additionally, our distributed-order fractional operator $\int D^\alpha \mathbf{X}(t)\mathrm{d}\mu(\alpha)$ acts as a flexible superposition of the dynamics driven by individual fractional-order operators $D^\alpha$. This approach allows for nuanced dynamics that adapt to diverse waiting times. Theorem 2 demonstrates its capability to approximate any waiting time distribution $f(n)$ for graph-based random walkers, thereby providing versatility in modeling feature updating dynamics with varied memory incorporation levels.

# F   Integer-Order Continuous GNNs

## F.1   GRAND and GraphCON

For the general GRAND model, the governing equation is given by:

$$\frac{\mathrm{d}\mathbf{X}(t)}{\mathrm{d}t} = (\mathbf{A}(\mathbf{X}(t)) - \mathbf{I})\mathbf{X}(t). \tag{38}$$

In the case of GRAND-l, the adjacency matrix $\mathbf{A}(\mathbf{X}(t))$ remains constant throughout the integration process, i.e., $\mathbf{A}(\mathbf{X}(t)) = \mathbf{A}(\mathbf{X}(0))$.

For GRAND-nl, the adjacency matrix $\mathbf{A}(\mathbf{X}(t))$ is time-varying and is calculated using $\mathbf{X}(t)$ with the attention mechanism. The entries of $\mathbf{A}(\mathbf{X}(t))$ are given by:

$$a(\mathbf{x}_i, \mathbf{x}_j) = \mathrm{softmax}\left(\frac{(\mathbf{W}_K \mathbf{x}_i)^\top \mathbf{W}_Q \mathbf{x}_j}{\bar{d}_k}\right), \tag{39}$$

where $\mathbf{W}_K$ and $\mathbf{W}_Q$ are learned matrices, and $\bar{d}_k$ is a hyperparameter determining the dimension of $\mathbf{W}_k$.

**GraphCON** [9]: Influenced by oscillator dynamical systems, GraphCON is given by the following second-order differential equation

$$\frac{\mathrm{d}^2\mathbf{X}(t)}{\mathrm{d}t^2} = \sigma(\mathbf{F}_\theta(\mathbf{X}(t), t)) - \gamma\mathbf{X}(t) - \beta\frac{\mathrm{d}\mathbf{X}(t)}{\mathrm{d}t}, \tag{40}$$

where $\mathbf{F}_\theta(\cdot)$ represents a learnable 1-neighborhood coupling function, $\sigma$ is an activation function, and $\gamma$ and $\beta$ are adjustable parameters. Equivalently, we have

$$\begin{cases} \frac{\mathrm{d}\mathbf{Y}(t)}{\mathrm{d}t} = \sigma(\mathbf{F}_\theta(\mathbf{X}(t), t)) - \gamma\mathbf{X}(t) - \beta\mathbf{Y}(t), \\ \frac{\mathrm{d}\mathbf{X}(t)}{\mathrm{d}t} = \mathbf{Y}(t), \end{cases} \tag{41}$$

In the case of GraphCON-l, similar to GRAND-l, $\mathbf{F}_\theta(\mathbf{X}(t), t) = \mathbf{A}(\mathbf{X}(t)) = \mathbf{A}(\mathbf{X}(0))$. For GraphCON-nl, similar to GRAND-nl, $\mathbf{F}_\theta(\mathbf{X}(t), t) = \mathbf{A}(\mathbf{X}(t))$, where $\mathbf{A}(\mathbf{X}(t))$ is still obtained from (39).

### F.2 Other Continuous GNNs

**Heterophilic CDE** [12]: Based on the convection-diffusion equation, Heterophilic CDE includes both a diffusion and convection term to address information propagation from heterophilic neighbors:

$$\frac{d\mathbf{X}(t)}{dt} = (\mathbf{A}(\mathbf{X}(t)) - \mathbf{I})\mathbf{X}(t) + \text{div}(\mathbf{V}(t) \circ \mathbf{X}(t)), \tag{42}$$

where $\mathbf{V}_{ij}(t) \in \mathbb{R}^d$ is the velocity vector associated with each edge $(i, j)$ at time $t$, $\mathbf{V}(t) = \{\mathbf{V}_{ij}(t)\}_{(i,j)\in\mathcal{E}}$, ($\mathcal{E}$ is the edge set containing all the pairs $(i, j)$ s.t. $W_{ij} \neq 0$) and

$$i\text{-th row of } (\text{div}(\mathbf{V}(t) \circ \mathbf{X}(t))) = \sum_{j:(i,j)\in\mathcal{E}} \mathbf{V}_{ij}(t) \odot \mathbf{x}_j(t) \tag{43}$$

for each node $i \in \mathcal{V}$. The velocity $\mathbf{V}_{ij}(t)$ is given by

$$\mathbf{V}_{ij}(t) = \sigma\left(\mathbf{M}(\mathbf{x}_j(t) - \mathbf{x}_i(t))\right), \tag{44}$$

with $\mathbf{M}$ is a learnable matrix and $\sigma$ denotes an activation function.

**GREAD:** To tackle the challenges associated with heterophilic graphs, the paper [11] introduces the GREAD model. This model extends the GRAND framework by incorporating a reaction term, thereby establishing a diffusion-reaction equation for GNNs. The governing equation for this model is expressed as:

$$\frac{d\mathbf{X}(t)}{dt} = -\alpha\mathbf{L}(\mathbf{X}(t)) + \beta r(\mathbf{X}(t)), \tag{45}$$

where $r(\mathbf{X}(t))$ is a reaction term, $\alpha$ and $\beta$ are trainable parameters designed to balance each term.

**GRAND++:** Building upon the GRAND model, the paper [8] presents the GRAND++ model. This enhancement adds a source term to the original GRAND framework, aimed at addressing challenges associated with training on limited labeled data. The differential equation used in GRAND++ is:

$$\frac{d\mathbf{X}(t)}{dt} = -\mathbf{L}(\mathbf{X}(t)) + \mathbf{C}(0) \tag{46}$$

where $\mathbf{L}(\mathbf{X}(t))$ denotes the graph Laplacian matrix, and $\mathbf{C}(0)$ represents a subset of $\mathbf{X}(0)$, consisting only of nodes identified as "trustworthy".

## G  Implementation Specifics and Dataset Details

### G.1  Example Details

The distributed-order fractional model is a natural generalization of single-term as well as multi-term fractional order models. It is more powerful and practical in applications. Due to multiscale characteristics in some physics problems, single-term fractional order model fails to capture this feature. Though multi-term fractional order models can capture multiscale properties, they are unsuitable in applications where the number of terms and corresponding fractional orders are unknown. However, the distributed-order fractional model is capable of dealing with multiscale characteristics and does not require knowing the number of terms and corresponding fractional orders a priori. Graph data has a complex nature as it is from the real world. Therefore, it is natural to use a distributed-order fractional model.

- **Kelvin-Voigt** model [33]:

$$\sigma(t) = \mathrm{E}\tau^\gamma \int_0^1 D^\alpha \epsilon(t) d\alpha.$$

- **Maxwell** model [31]:

$$\sigma(t) = \mathrm{E}_\infty \tau^\alpha D^\alpha \epsilon(t).$$

- **Zener** model [32]:

$$(1 + \frac{a_o}{b_o})\sigma(t) = a_o D^\alpha \epsilon(t) + c_o(1 + \frac{a_o}{b_o})D^\beta \epsilon(t).$$

For simplicity, take $E = E_\infty = 1, \tau = 1, a_o = b_o = 0.1, c_o = 1/4$ and $\alpha = 0.3$ in Maxwell model, $\alpha = 0.2, \beta = 0.6$ in Zener model. The toy data is generated for a common $\sigma(t) = \cos(t)$ and $\epsilon(0) = 0.5$. We generate the data through an open source package FractionalD-iffEq.jl (https://scifracx.org/FractionalDiffEq.jl/stable/) that is totally driven by Julia and licensed with MIT License. We follow standard setups and apply built-in algorithms. Specifically, we choose PIEX algorithm which is an explicit method for Maxwell and Zener models, and DOMatrixDiscrete algorithm that is a strip matrix method for Kelvin-Voigt model.

For the implementation of FROND-NN and DRAGON-NN, we split the data into 80% training and 20% testing sets. We construct identical two-layer neural networks with activation functions for both FROND and DRAGON. Using the current observations, we predict the next 10 points in the trajectory on the test data and calculate the MSE.

### G.2 Dataset Details

In this subsection, we detail the statistics of the datasets utilized in this paper, as illustrated in Tables 5 to 7. The datasets span various domains and scales, providing a comprehensive evaluation of DRAGON's performance.

Table 5: Dataset statistics used in Table 3

| Dataset | Type | Classes | Features | Nodes | Edges |
|---------|------|---------|----------|-------|-------|
| Cora | citation | 7 | 1433 | 2485 | 5069 |
| Citeseer | citation | 6 | 3703 | 2120 | 3679 |
| PubMed | citation | 3 | 500 | 19717 | 44324 |
| Coauthor CS | co-author | 15 | 6805 | 18333 | 81894 |
| Computers | co-purchase | 10 | 767 | 13381 | 245778 |
| Photos | co-purchase | 8 | 745 | 7487 | 119043 |
| CoauthorPhy | co-author | 5 | 8415 | 34493 | 247962 |
| Airport | tree-like | 4 | 4 | 3188 | 3188 |
| Disease | tree-like | 2 | 1000 | 1044 | 1043 |

Table 6: Dataset statistics of used in Table 4

| Dataset | Nodes | Edges | Classes | Node Features |
|---------|-------|-------|---------|---------------|
| Roman-empire | 22662 | 32927 | 18 | 300 |
| Wiki-cooc | 10000 | 2243042 | 5 | 100 |
| Minesweeper | 10000 | 39402 | 2 | 7 |
| Questions | 48921 | 153540 | 2 | 301 |
| Workers | 11758 | 519000 | 2 | 10 |
| Amaon-ratings | 24492 | 93050 | 5 | 300 |

Table 7: Dataset and graph statistics used in Table 10

| Dataset | Graphs (Fake) | Total Nodes | Total Edges | Avg. Nodes per Graph |
|---------|---------------|-------------|-------------|----------------------|
| Politifact (POL) | 314 (157) | 41,054 | 40,740 | 131 |
| Gossipcop (GOS) | 5464 (2732) | 314,262 | 308,798 | 58 |

## H  Time Complexity

In this section, we discuss the time complexity of the model, as detailed in Table 8 and Table 9. It is observed that the DRAGON framework exhibits computational costs comparable to those of

traditional continuous GNN models. All experiments are conducted on NVIDIA GeForce RTX 3090 or A5000 GPUs with 24GB of memory.

Table 8: Inference time of models on the Cora dataset: integral time $T = 10$ and step size of 1

| Model | D-GRAND-l | D-GRAND-nl | D-GraphCON-l | D-GraphCON-nl | D-CDE |
|---|---|---|---|---|---|
| Inf. Time(ms) | 3.78 | 7.21 | 4.18 | 7.80 | 13.68 |
| Model | F-GRAND-l | F-GRAND-nl | F-GraphCON-l | F-GraphCON-nl | F-CDE |
| Inf. Time(ms) | 3.29 | 6.62 | 4.15 | 7.37 | 13.18 |
| Model | GRAND-l | GRAND-nl | GraphCON-l | GraphCON-nl | CDE |
| Inf. Time(ms) | 2.06 | 5.33 | 3.32 | 6.86 | 12.23 |

Table 9: Training time per epoch on the Cora dataset: integral time $T = 10$ and step size of 1

| Model | D-GRAND-l | D-GRAND-nl | D-GraphCON-l | D-GraphCON-nl | D-CDE |
|---|---|---|---|---|---|
| Train. Time(ms) | 30.93 | 78.33 | 40.77 | 82.52 | 160.20 |
| Model | F-GRAND-l | F-GRAND-nl | F-GraphCON-l | F-GraphCON-nl | F-CDE |
| Train. Time(ms) | 29.76 | 70.31 | 37.82 | 73.10 | 148.92 |
| Model | GRAND-l | GRAND-nl | GraphCON-l | GraphCON-nl | CDE |
| Train. Time(ms) | 22.17 | 74.39 | 41.23 | 88.83 | 166.48 |

# I More Experiment Results

## I.1 Graph Classification

Following the experiments of FROND [15], we perform graph classification tasks on the FakeNews-Net datasets [75]. The dataset features a diverse array of node features, including BERT embeddings, features derived from spaCy's pre-trained models, and profile-specific features from Twitter accounts. The performance outcomes, as detailed in Table 10, reveal that the DRAGON-based model outperforms its counterparts, showcasing the significant enhancements brought about by the DRAGON framework. This is because DRAGON enables feature updating dynamics with flexible memory effects stemming from the coexistence of multiple orders of derivatives.

Table 10: Graph classification results

| Feature | POL | | | GOS | | |
|---|---|---|---|---|---|---|
| | Profile | word2vec | BERT | Profile | word2vec | BERT |
| GraphSage | 77.60±0.68 | 80.36±0.68 | 81.22±4.81 | 92.10±0.08 | 96.58±0.22 | 97.07±0.23 |
| GCN | 78.28±0.52 | 83.89±0.53 | 83.44±0.38 | 89.53±0.49 | 96.28±0.08 | 95.96±0.75 |
| GAT | 74.03±0.53 | 78.69±0.78 | 82.71±0.19 | 91.18±0.23 | 96.57±0.34 | 96.61±0.45 |
| GRAND-l | 77.83±0.37 | 86.57±1.13 | 85.97±0.74 | 96.11±0.26 | 97.04±0.55 | 96.77±0.34 |
| F-GRAND-l | **79.49±0.43** | **88.69±0.37** | **89.29±0.93** | **96.40±0.19** | **97.40±0.03** | **97.53±0.14** |
| D-GRAND-l | **79.58±0.37** | **88.94±0.35** | **89.44±0.56** | **97.14±0.32** | **97.62±0.06** | **97.83±0.17** |

## I.2 Oversmoothing Mitigation

The FROND framework has demonstrated strong performance in mitigating the oversmoothing issue in GNNs [15]. As shown in Theorem 2, DRAGON can approximate any waiting time distribution, suggesting its potential to address the oversmoothing problem as well. To verify this, we conduct node classification experiments under different integration times, which can be viewed as the number of layers when the step size is set to 1. From Table 11, we observe that the DRAGON framework maintains comparable performance across various depths, demonstrating consistent mitigation of the

oversmoothing issue. Furthermore, we find that DRAGON obviously outperforms FROND on the Pubmed dataset.

Table 11: Oversmoothing mitigation under fixed data splitting without using largest connected component (LCC). '-' indicates the numerical solvers failed.

| Dataset | Model | 4 | 8 | 16 | 32 | 64 | 128 |
|---|---|---|---|---|---|---|---|
| Cora | GCN | 81.35±1.27 | 15.30±3.63 | 19.70±7.06 | 21.86±6.09 | 13.0±0.00 | 13.00±0.00 |
| | GAT | 80.95±2.28 | 31.90±0.00 | 31.90±0.00 | 31.90±0.00 | 31.90±0.00 | 31.90±0.00 |
| | GRAND-l | 81.29±0.43 | 81.50±0.87 | 80.58±0.63 | 79.80±0.56 | 79.10±0.62 | 73.80±0.82 |
| | F-GRAND-l | 81.17±0.75 | 82.68±0.64 | 82.24±1.17 | 81.43±1.01 | 81.33±0.88 | 80.60±0.98 |
| | D-GRAND-l | 81.02±0.76 | 82.92±0.78 | 82.82±0.78 | 82.28±0.91 | 81.62±0.76 | 81.17±0.74 |
| Citeseer | GCN | 68.84±2.46 | 61.58±2.09 | 10.64±1.79 | 7.70±0.00 | 7.70±0.00 | 7.70±0.00 |
| | GAT | 65.20±0.57 | 18.10±0.00 | 18.10±0.00 | 18.10±0.00 | 18.10±0.00 | 18.10±0.00 |
| | GRAND-l | 70.72±1.10 | 70.39±0.68 | 70.52±0.74 | 68.90±1.50 | 68.01±1.47 | 63.45±2.86 |
| | F-GRAND-l | 70.68±1.23 | 70.70±1.56 | 71.14±1.22 | 70.85±0.57 | 70.50±0.84 | 70.00±0.60 |
| | D-GRAND-l | 71.46±0.87 | 71.66±0.43 | 71.50±0.58 | 71.38±1.06 | 71.17±1.35 | 70.97±0.90 |
| Pubmed | GCN | 76.44±1.52 | 72.66±2.84 | 39.52±1.60 | 40.10±2.04 | 38.40±1.34 | 38.42±1.87 |
| | GAT | 76.98±1.23 | 40.70±0.00 | 40.70±0.00 | 40.70±0.00 | 40.70±0.00 | 40.70±0.00 |
| | GRAND-l | 77.94±0.24 | 78.22±0.70 | 77.84±0.54 | – | – | – |
| | F-GRAND-l | 78.96±0.64 | 79.08±0.61 | 79.62±0.47 | 79.04±0.74 | 78.60±0.68 | 74.60±0.73 |
| | D-GRAND-l | 78.42±0.13 | 78.72±0.30 | 78.80±0.82 | 78.56±0.62 | 79.28±0.26 | 79.50±0.55 |

## I.3 D-GREAD

Building upon the GREAD model [11], we introduce D-GREAD with the following formulation:

$$\int_0^1 D^\alpha \mathbf{X}(t) \, \mathrm{d}\mu(\alpha) = -\alpha \mathbf{L}(\mathbf{X}(t)) + \alpha r(\mathbf{X}(t)) \tag{47}$$

Following the experimental setting in [11], we conduct a node classification task on three heterophilic graph datasets, adhering to the data split method described in [76]. The baseline results are directly reported from [11]. As shown in Table 12, the DRAGON framework significantly improves upon the corresponding continuous GNNs, achieving the best performance across all three datasets. Notably, even the GRAND model, which traditionally underperforms on heterophilic graph datasets, performs exceptionally well when integrated with the DRAGON framework. This demonstrates the DRAGON framework's capability to learn a wide range of temporal dynamics and seamlessly integrate with continuous GNNs.

Table 12: Node classification results (%) of heterophilic graph under fixed data splits [76]

| Dataset | Texas | Wisconsin | Cornell |
|---|---|---|---|
| Geom-GCN [76] | 66.76±2.72 | 64.51±3.66 | 60.54±3.67 |
| H2GCN [57] | 84.86±7.23 | 87.65±4.98 | 82.70±5.28 |
| GGCN [77] | 84.86±4.55 | 86.86±3.29 | 85.68±6.63 |
| LINKX [78] | 74.60±8.37 | 75.49±5.72 | 77.84±5.81 |
| GloGNN [60] | 84.32±4.15 | 87.06±3.53 | 83.51±4.26 |
| ACM-GCN [63] | 87.84±4.40 | 88.43±3.22 | 85.14±6.07 |
| GCNII [79] | 77.57±3.83 | 80.39±3.40 | 77.86±3.79 |
| CGNN [6] | 71.35±4.05 | 74.31±7.26 | 66.22±7.69 |
| GRAND [7] | 75.68±7.25 | 79.41±3.64 | 82.16±7.09 |
| BLEND [80] | 83.24±4.65 | 84.12±3.56 | 85.95±6.82 |
| Sheaf [64] | 85.05±5.51 | 89.41±4.74 | 84.86±4.71 |
| GRAFF [81] | 88.38±4.53 | 87.45±2.94 | 83.24±6.49 |
| GREAD [11] | 88.92±3.72 | 89.41±3.30 | 86.49±7.15 |
| F-GREAD | 89.46±3.74 | 89.57±3.36 | 86.89±4.16 |
| D-GRAND | 86.49±3.20 | 90.39±3.97 | **90.0±4.67** |
| D-GREAD | **90.54±3.25** | **90.98±3.30** | 89.46±4.26 |

## I.4 D-GRAND++

Expanding on the GRAND++ model [8], we introduce D-GRAND++ with the following formulation:

$$\int_0^1 D^\alpha \mathbf{X}(t) \, \mathrm{d}\mu(\alpha) = -\mathbf{L}(\mathbf{X}(t)) + \mathbf{C}(0) \tag{48}$$

We adhere to the experimental framework outlined in the GRAND++ study, focusing specifically on the model's efficacy in limited-label scenarios. The key difference in our approach is the integration of DRAGON framework. Our results in Table 13 clearly show that D-GRAND++ not only consistently outperforms the baseline GRAND++ across various tests but also shows competitive performance with F-GRAND++.

Table 13: Node classification results (%) under limited-label scenarios

| Model | pre class | Cora | Citeseer | Pubmed | CoauthorCS | Computer | Photo |
|---|---|---|---|---|---|---|---|
| GRAND++ | 1 | 54.94±16.09 | 58.95±9.59 | 65.94±4.87 | 60.30±1.50 | 67.65±0.37 | 83.12±0.78 |
| F-GRAND++ | 1 | **57.31±8.89** | 59.11±6.73 | **65.98±2.72** | 67.71±1.91 | 67.65±0.37 | **83.12±0.78** |
| D-GRAND++ | 1 | 55.84±8.79 | **60.0±8.22** | 65.80±2.88 | **69.59±3.81** | **67.84±0.21** | 83.00±0.64 |
| GRAND++ | 2 | 66.92±10.04 | 64.98±8.31 | 69.31±4.87 | 76.53±1.85 | 74.47±1.48 | 83.71±0.90 |
| F-GRAND++ | 2 | 70.09±8.36 | **64.98±8.31** | 69.37±5.36 | 77.97±2.35 | 78.85±0.96 | **83.71±0.90** |
| D-GRAND++ | 2 | **71.21±8.27** | 62.10±6.83 | **69.97±5.28** | **82.24±2.59** | **79.15±0.82** | 83.59±1.24 |
| GRAND++ | 5 | 77.80±4.46 | 70.03±3.63 | 71.99±1.91 | 84.83±0.84 | 82.64±0.56 | 88.33±1.21 |
| F-GRAND++ | 5 | 78.79±1.66 | 70.26±2.36 | 73.38±5.67 | 86.09±2.09 | **82.64±0.56** | 88.56±0.67 |
| D-GRAND++ | 5 | **79.18±1.22** | **70.83±3.98** | **73.57±2.85** | **88.46±0.95** | 82.23±0.78 | **88.99±0.71** |
| GRAND++ | 10 | 80.86±2.99 | 72.34±2.42 | 75.13±3.88 | 86.94±0.46 | 82.99±0.81 | 90.65±1.19 |
| F-GRAND++ | 10 | 82.73±0.81 | 73.52±1.44 | 77.15±2.87 | 87.85±1.44 | 83.26±0.41 | 91.15±0.52 |
| D-GRAND++ | 10 | **82.94±1.32** | **74.18±0.40** | **77.63±3.08** | **89.52±0.35** | **83.65±0.94** | **91.37±0.51** |
| GRAND++ | 20 | 82.95±1.37 | 73.53±3.31 | 79.16±1.37 | 90.80±0.34 | 85.73±0.50 | 93.55±0.38 |
| F-GRAND++ | 20 | **84.57±1.07** | **74.81±1.78** | **79.96±1.68** | 91.03±0.72 | **85.78±0.43** | 93.55±0.38 |
| D-GRAND++ | 20 | 84.41±0.96 | 73.99±2.60 | 79.39±1.42 | **91.98±0.33** | 85.81±0.69 | 93.28±0.30 |

## I.5  D(oscillation)-GRAND

Our framework accommodates any floating value for $\alpha$. Nonetheless, for the experiments presented in our main paper, we have specified $\alpha \in [0, 1]$ to ensure a fair comparison by maintaining identical initial conditions to those utilized in the original models.

For instance, we can let $\alpha$ range between 0 and 2, leading to the D(oscillation)-GRAND model:

GRAND: $\frac{\mathrm{d}\mathbf{X}(t)}{\mathrm{d}t} = (\mathbf{A}(\mathbf{X}(t)) - \mathbf{I})\mathbf{X}(t)$

D-GRAND: $\int_0^1 D^\alpha \mathbf{X}(t)\mathrm{d}\mu(\alpha) = (\mathbf{A}(\mathbf{X}(t)) - \mathbf{I})\mathbf{X}(t)$

D(oscillation)-GRAND: $\int_0^2 D^\alpha \mathbf{X}(t)\mathrm{d}\mu(\alpha) = (\mathbf{A}(\mathbf{X}(t)) - \mathbf{I})\mathbf{X}(t)$

In contrast to GRAND and D-GRAND, which employ the initial condition $\mathbf{X}(0) = \mathbf{X}$, D(oscillation)-GRAND is characterized as an oscillation-type differential equation and adopts the initial condition $\mathbf{X}'(0) = \mathbf{X}(0) = \mathbf{X}$. However, comparing this model to GRAND or F-GRAND makes it challenging to ascertain whether performance differences arise from the varied initial conditions or the incorporation of distributed fractional derivatives. To preserve the D-GRAND as a diffusion-type equation with the same initial condition as its counterparts, GRAND and F-GRAND, we limit $\alpha$ to the range $0 < \alpha \le 1$.

We showcase preliminary results for D(oscillation)-GRAND in Table 14. The findings reveal that D(oscillation)-GRAND does not outperform GRAND or D-GRAND on these datasets, suggesting that increasing the value of $\alpha$ does not contribute positively to these tasks and instead elevates the model's complexity.

Table 14: Comparison between GRAND, D-GRAND, and D(oscillation)-GRAND

| | Cora | Citeseer | Pubmed | Airport | Disease |
|---|---|---|---|---|---|
| GRAND | 83.6±1.0 | 73.4±0.5 | 78.8±1.7 | 80.5±9.6 | 74.5±3.4 |
| D-GRAND | **85.1±1.3** | **74.5±1.1** | **79.6±2.6** | **98.5±0.1** | **93.2±2.5** |
| D(oscillation)-GRAND | 82.6±1.6 | 72.8±1.8 | 78.1±2.3 | 93.5±0.6 | 89.5±2.4 |

### I.6 Sensitivity Analysis

As demonstrated in our main paper, a significant advantage of the DRAGON framework is its ability to learn the optimal $\alpha$ through the adjustment of weights $w_j$ in (14). We analyze the impact of varying the number $n$ in (14) on the final accuracy. The findings, illustrated in Table 15 and Table 16, reveal that test accuracy remains stable despite changes in $n$, underscoring DRAGON's robustness against parameter selection. This stability highlights the framework's considerable improvements, as compared with the FROND framework results depicted in Fig. 2.

Table 15: Learned $w_j$ of Airport dataset

| 1.0 | 0.9 | 0.8 | 0.7 | 0.6 | 0.5 | 0.4 | 0.3 | 0.2 | 0.1 | Accuracy |
|-----|-----|-----|-----|-----|-----|-----|-----|-----|-----|----------|
| | | | | | $\alpha_j$ | | | | | |
| 1.61 | 1.53 | 1.44 | 1.34 | 1.22 | 1.07 | 0.90 | 0.63 | 0.39 | 0.07 | 98.50±0.13 |
| 1.59 | × | 1.43 | × | 1.22 | × | 0.94 | × | 0.48 | × | 98.38±0.15 |
| × | 2.15 | × | 1.31 | × | 0.58 | × | 0.17 | × | 0.02 | 97.70±0.54 |
| 1.61 | 1.29 | × | × | × | × | × | -0.01 | × | -0.03 | 98.06±0.39 |
| × | 1.85 | × | × | 0.95 | × | 0.35 | × | × | × | 98.38±0.15 |
| × | × | 3.09 | × | × | × | 1.69 | × | 0.87 | × | 98.12±0.44 |
| × | × | × | × | × | 3.58 | × | 2.32 | × | 0.84 | 98.12±0.44 |

Table 16: Learned $w_j$ of Roman-empire dataset.

| 1.0 | 0.9 | 0.8 | 0.7 | 0.6 | 0.5 | 0.4 | 0.3 | 0.2 | 0.1 | Accuracy |
|-----|-----|-----|-----|-----|-----|-----|-----|-----|-----|----------|
| | | | | | $\alpha_j$ | | | | | |
| 1.62 | 1.09 | 0.59 | 0.16 | -0.20 | -0.46 | -0.62 | -0.68 | -0.62 | -0.38 | 93.87±0.41 |
| 1.30 | × | 0.43 | × | -0.26 | × | -0.70 | × | -0.77 | × | 93.52±0.40 |
| × | 1.29 | × | 0.46 | × | -0.21 | × | -0.64 | × | -0.59 | 93.50±0.42 |
| 0.58 | 0.45 | × | × | × | × | × | -0.07 | -0.09 | × | 93.10±0.33 |
| × | 0.64 | × | × | 0.31 | × | 0.12 | × | × | × | 93.22±0.36 |
| × | × | 0.73 | × | × | × | 0.23 | × | 0.06 | × | 93.09±0.46 |
| × | × | × | × | × | 0.67 | × | 0.34 | × | 0.08 | 93.09±0.25 |

### I.7 Large Scale Ogb-Products dataset

To showcase the scalability of the DRAGON framework to large-scale datasets, we expand our evaluation to include the Ogb-products dataset, following the experimental protocols detailed in [82]. To manage this extensive dataset effectively, we adopted a mini-batch training strategy that involves sampling nodes and constructing subgraphs, as introduced by GraphSAINT [83]. The outcomes presented in Table 17 demonstrate that the DRAGON-based model outperforms others, highlighting DRAGON's efficiency and scalability.

### I.8 Hyperparameters

The hyperparameters employed in Table 4 are detailed in Table 18. For the hyperparameters pertaining to all other experiments, they will be disclosed alongside the code release.

## J  Proofs of Results

In this section, we provide detailed proofs of the results stated in the main paper.

Table 17: Node classification accuracy(%) on Ogb-products dataset

| Model | MLP | Node2vec | Full-batch GCN | GraphSAGE | GRAND-l | F-GRAND-l | D-GRAND-l |
|-------|-----|----------|----------------|-----------|---------|-----------|-----------|
| Acc | 61.06±0.08 | 72.49±0.10 | 75.64±0.21 | 78.29±0.16 | 75.56±0.67 | 76.61±0.78 | **78.81±0.19** |

Table 18: Hyper-parameters used in Table 4

| Dataset | Model | lr | weight decay | indrop | dropout | hidden dim | time | step size |
|---------|-------|-----|-----|-----|-----|-----|-----|-----|
| Roman-empire | D-CDE | 0.005 | 0.0001 | 0.4 | 0.2 | 80 | 4 | 0.2 |
| Wiki-cooc | D-CDE | 0.001 | 0.0001 | 0.4 | 0.4 | 64 | 4 | 1 |
| Minesweeper | D-CDE | 0.005 | 0.0001 | 0.2 | 0.4 | 64 | 4 | 0.2 |
| Questions | D-CDE | 0.005 | 0.0001 | 0.4 | 0.4 | 16 | 8 | 1 |
| Workers | D-CDE | 0.005 | 0.0001 | 0 | 0.4 | 64 | 3 | 0.5 |
| Amazon-ratings | D-CDE | 0.001 | 0.0001 | 0.2 | 0.4 | 128 | 4 | 0.2 |

## J.1 Proof of Theorem 1

*Proof.* Noting that $\sum_{n=1}^{\infty} \psi_{\alpha_o}(n) = 1$, we subtract $\sum_{n=1}^{\infty} \psi_{\alpha_o}(n)\mathbb{P}_j(t - n\Delta\tau; \alpha_o)$ from both sides of (12) to yield the following:

$$\sum_{n=1}^{\infty} \left(\mathbb{P}_j(t; \alpha_o) - \mathbb{P}_j(t - n\Delta\tau; \alpha_o)\right) \psi_{\alpha_0}(n)$$

$$= (\Delta\tau)^{\alpha_o} d_{\alpha_o} |\Gamma(-\alpha_o)| \sum_{n=1}^{\infty} \left[ \sum_{\substack{i \in \mathcal{V} \\ i \neq j}} \mathbb{P}_i(t - n\Delta\tau; \alpha_o)\frac{W_{ij}}{d_i} - \mathbb{P}_j(t - n\Delta\tau; \alpha_o) \right] \psi_{\alpha_o}(n)$$

$$= (\Delta\tau)^{\alpha_o} d_{\alpha_o} |\Gamma(-\alpha_o)| \sum_{n=1}^{\infty} \left[\mathbf{L}\mathbb{P}(t - n\Delta\tau; \alpha_o)\right]_j \psi_{\alpha_o}(n).$$

Divide both sides by $(\Delta\tau)^{\alpha_o} d_{\alpha_o} |\Gamma(-\alpha_o)|$, we have

$$\frac{1}{|\Gamma(-\alpha_o)|} \sum_{n=1}^{\infty} \frac{\mathbb{P}_j(t; \alpha_o) - \mathbb{P}_j(t - n\Delta\tau; \alpha_o)}{(n\Delta\tau)^{1+\alpha_o}} \Delta\tau$$

$$= \sum_{n=1}^{\infty} \left[\mathbf{L}\mathbb{P}(t - n\Delta\tau; \alpha_o)\right]_j \psi_{\alpha_o}(n).$$

Let $\Delta\tau \to 0$ and switch the limit and the summation according to dominated convergence theorem (we assume the conditions are satisfied), we have

$$\frac{1}{|\Gamma(-\alpha_o)|} \int_0^{\infty} \frac{\mathbb{P}_j(t; \alpha_o) - \mathbb{P}_j(t - \tau; \alpha_o)}{\tau^{1+\alpha_o}} \, d\tau$$

$$= \left[\mathbf{L}\mathbb{P}(t; \alpha_o)\right]_j.$$

Since $\Gamma(1 - \alpha_o) = \alpha_o \Gamma(-\alpha_o)$, according to (2), we have

$$_M D^{\alpha_o} \mathbb{P}(t; \alpha_o) = \mathbf{L}\mathbb{P}(t; \alpha_o).$$

The proof is now complete. $\square$

## J.2 Proof of Theorem 2

*Proof.* Let $r_i = \alpha_i + 1$. It is obvious that $\sum_{i \geq 1} 1/r_i = \infty$. Let $C[0, 1]$ be the space of continuous function on the interval $[0, 1]$ with the $\infty$-norm. By the Müntz–Szász theorem [84], the span of $\{x^{r_i}, r_i \in \mathbb{R}\}$ is dense in $C[0, 1]$.

Consider any $f \in C_0(\mathbb{N})$. We define a function $\overline{f} \in C[0, 1]$ associated with $f$ as follows. We set $\overline{f}(0) = 0, \overline{f}(1/n) = f(n), n \in \mathbb{N}$. We then linearly interpolate between $1/n + 1$ and $1/n$ for any $n \geq 1$ to obtain $\overline{f}$ on the remaining points of $[0, 1]$. Apart from 0, the function $\overline{f}$ is piecewise linear and hence continuous. It is also continuous at 0 as $f$ is vanishing at $\infty$.

According to the first paragraph, for any $\epsilon > 0$, we can find a $N$ and coefficients $\{w_i, 0 \leq i \leq N\}$ such that

$$\left| \overline{f}(x) - \sum_{i=0}^{N} w_i x^{r_i} \right| < \epsilon, \text{ for any } x \in [0, 1].$$

Letting $x = 0$, we see that $|w_0| < \epsilon$. Therefore, for any $n \in \mathbb{N}$, we have

$$
\left| f(n) - \sum_{i=1}^{N} w_i' \cdot \psi_{\alpha_i}(n) \right|
$$

$$
= \left| \overline{f}(\frac{1}{n}) - \sum_{i=1}^{N} w_i \cdot \frac{1}{n^{r_i}} \right|
$$

$$
\leq \left| \overline{f}(\frac{1}{n}) - \sum_{i=0}^{N} w_i \cdot \frac{1}{n^{r_i}} \right| + \epsilon
$$

$$
\leq 2\epsilon.
$$

where $w_i'$ is defined s.t. $w_i = w_i' d_{\alpha_i}$. The proof is now complete[3]. □

## K    Limitations and Broader Impacts

This paper proposes a generalized framework, DRAGON, that enhances existing continuous GNNs. However, its application is currently limited to continuous GNNs. For other types of GNNs, such as graph transformers [85], they need to be transformed into the formulation of differential equations before being combined with DRAGON. A future direction to address this limitation is to develop a more general DRAGON framework that does not rely on numerical solvers. Regarding broader impacts, the future societal impact of this work depends on a commitment to ethical standards and responsible use. It is crucial to ensure that advancements lead to positive outcomes without compromising individual rights or contributing to inequality.

## L    Contribution Statement

The concept of DRAGON was initially proposed by Feng Ji and the framework is fully developed by Qiyu Kang and Kai Zhao. The manuscript was written collaboratively by Kai Zhao, Xuhao Li, and Qiyu Kang. Theoretical support for FDE was provided by Feng Ji, Qiyu Kang, Xuhao Li, and Qinxu Ding. Kai Zhao was responsible for writing the implementation code and organizing the experiments. Wenfei Liang and Yanan Zhao provided experimental support. Guidance throughout the process was provided by Wee Peng Tay.

---

[3]The proof is based on the answers to a question posted by F. Ji on MathOverflow `https://mathoverflow.net/questions/446194/stone-weierstrass-without-the-subalgebra-condition/446221#446221`

