# OpenReview forum: "Distributed-Order Fractional Graph Operating Network"
_NeurIPS.cc/2024/Conference — NeurIPS 2024 spotlight_

### Official Review · Reviewer_smsh · 2024-07-12

**Soundness:** 3
**Presentation:** 2
**Contribution:** 3
**Rating:** 5
**Confidence:** 3

**Summary:**

This paper introduces a continuous Graph Neural Network (GNN) framework DRAGON that leverages distributed-order fractional calculus. Unlike traditional continuous GNNs that utilize integer-order or single fractional-order differential equations, DRAGON employs a learnable probability distribution over a range of real numbers for the derivative orders. This approach allows for a flexible superposition of multiple derivative orders, enabling the model to capture complex graph feature updating dynamics. provide a non-Markovian graph random walk interpretation under a specific anomalous diffusion setting. The authors conduct extensive experiments across various graph learning tasks and achieve competitive results.

**Strengths:**

The use of distributed-order fractional calculus in GNNs is novel to me,  and expands the capabilities of continuous GNN models.
They extend previous dynamical systems to operate under DRAGON framework and generalizes the scenarios to involve distributed-order fractional derivatives
The authors conduct experiments on different tasks and plenty o datasets and conduct sensitivity analysis to better evaluate their proposed framework

**Weaknesses:**

1. Can you provide more visualizations and explanations for non-Markovian graph random walk interpretation?
2. lack of code for reviewing and reproducibility
3. The paper does not thoroughly address the scalability of the proposed framework in real-world applications with very large graphs.

**Questions:**

1. How does the computational complexity and memory usage of DRAGON compare to other continuous GNN models in practice, especially for very large graphs? I have seen the time complexity comparison on Cora, but are there more experiments on a larger dataset since cora is small.
2. Can the authors provide their code in order for better reviewing?
3. in table 4, the performance of D-CDE is comparable to F-CDE. the improvements are not obvious.
4. Can the authors provide more explanations for the intrinsic property 'its ability to capture flexible memory effects' mentioned in the paper?

**Limitations:**

The authors discuss limitations in the Appendix I, like limitation to continuous GNNs. However, it is better that the authors have a discussion in the main context. Besides, the scalability and computational efficiency of the proposed framework in large-scale applications may need further discussion.

---

> ### Author Rebuttal · Authors · 2024-08-07
>
> ## Weakness 1 & Question 4:  Explanation of non-Markovian graph random walk with flexible memory
>
> **Response:**   Thank you for your valuable comments and suggestions.
>
> **1. Further explanation for non-Markovian graph random walk with flexible memory:**
> Lines 211-223 of the manuscript detail the dynamics of the random walk.
> For enhanced clarity, here we include the corresponding transition probability $\mathbb{P}$ representation for the non-Markovian random walker at time $t$, which explicitly accounts for node positions throughout the entire path history $(\ldots, \mathbf{R}(t-n \Delta \tau), \ldots, \mathbf{R}(t-\Delta \tau))$, as shown in Eq.(S4). Here, $\mathbf{R}\left(t\right)$ represents the walker's position on the graph nodes $\\{\mathbf{x}_j\\}\_{j=1}^{|\mathcal{V}|}$ at time $t$. This model ensures that all historical states influence transitions, emphasizing the model's non-Markovian nature. We consider a random walker navigating over graph $\mathcal{G}$ with an infinitesimal interval of time $\Delta \tau>0$. We assume that there is no self-loop in the graph topology. The transition probability of the random walker is characterized as follows:
>
>
> \begin{align}
> \begin{aligned}
> & \mathbb{P}\left(\mathbf{R}(t)=\mathbf{x}\_{j_t} \mid \ldots, \mathbf{R}(t-n \Delta \tau)=\mathbf{x}\_{j_{t-n \Delta \tau}}, \ldots, \mathbf{R}(t-\Delta \tau)=\mathbf{x}\_{j_{t-\Delta \tau}}\right) \\\\
> & = \begin{cases}
> \left(1- K \right)\psi_\alpha(n) & \text{if revisiting historical positions $\mathbf{R}(t-n\Delta \tau)$ with $j_{t}=j_{t-n\Delta \tau}$, i.e., the walker's wait time is $n\Delta \tau$ and stays at the same node,} \\\\
> \left(K\frac{W_{j_{t-\Delta \tau} j_{t}}}{d_{j_{t-\Delta \tau}}}\right)\psi_\alpha(n)  & \text{if jumping from historical positions $j_{t-n\Delta \tau}$ to $j_{t}$, i.e., the walker's wait time is $n\Delta \tau$ and jumps to $j_{t-n\Delta \tau}$'s neighbour $j_{t}$}.
> \end{cases} \quad  \quad  \quad   \quad \quad \text{(S4)}
> \end{aligned}
> \end{align}
> where $K\coloneqq(\Delta \tau)^\alpha d_\alpha|\Gamma(-\alpha)|$ is a normalization coefficient, $j_{t-n\Delta \tau}$ is the node index visited at time $t-n\Delta \tau$, and $\psi_\alpha(n)$ is the probability that the walker's waiting time is $n\Delta\tau$.
> For a specific $\alpha$, the waiting time $\psi_\alpha(n)$ follows a power-law distribution $\propto n^{-\left(\alpha+1\right)}$. Additionally, our distributed-order fractional operator $\int D^\alpha \mathbf{X}(t) \mathrm{d} \mu(\alpha)$ acts as a flexible superposition of the dynamics driven by individual fractional-order operators $D^\alpha$. This approach allows for nuanced dynamics that adapt to diverse waiting times. Theorem 2 in our DRAGON framework demonstrates its capability to approximate any waiting time distribution $f(n)$ for graph-based random walkers, thereby providing versatility in modeling feature updating dynamics with varied memory incorporation levels.
>
> **2. Visualization of the random walk:**
>
> We kindly refer the reviewer to the attached one-page PDF for the visualization.
>
> ## Weakness 2 & Question 2:  Missing implementation code (already included in our initial submission)
>
> **Response:** We would like to clarify that the implementation code has already been included in "Supplementary Material" of our initial submission.
>
> ## Weakness 3 & Question 1:  Scalability of DRAGON
>
> **Response:** Thank you for your comments. We would like to clarify that our manuscript includes experimental results on the ogbn-products dataset, which is a large-scale graph with 2,449,029 nodes and 61,859,140 edges, as presented in **Table 17** of our manuscript.
>
> Regarding the scalability of our DRAGON framework, we introduce two numerical solvers, detailed in Eq.(32) and (36). For further details on error analysis and complexity, we kindly refer the reviewer to **our Response to Weaknesses 2 for Reviewer uHnh** and the **model complexity** at the top **global response section**.
>
> Additionally, in the rebuttal, we have conducted a computational complexity comparison for the ogbn-arxiv dataset in **Table R1**. These results demonstrate that while our framework slightly increases computational costs compared to baseline continuous GNN models, it remains feasible for large graph datasets applications.
>
> ## Question 3:  Performance of D-CDE
>
> **Response:**
>
> Regarding classification performance, we acknowledge in Table 4 that our D-CDE marginally outperforms F-CDE. However, it is crucial to emphasize a key advantage: in the baseline FROND framework, the fractional derivative order $\alpha$ is a hyperparameter that requires tuning for each dataset. This tuning is effort-intensive and must be repeated to identify the optimal $\alpha$ for each new dataset. As shown by our results in Fig.1. of the manuscript, performance significantly deteriorates with non-optimal $\alpha$ values, highlighting FROND's sensitivity to this parameter.
>
> In contrast, our DRAGON framework utilizes a set of fractional orders $\alpha_j$ and learns their weights $w_j$ automatically, removing the need for manual tuning of $\alpha$. This method provides a more practical and robust solution compared to FROND, especially in scenarios where manual hyperparameter tuning is impractical or costly.

---

> > ### Comment · Reviewer_smsh · 2024-08-11
> >
> > Thank you for your code, responses and additional experiments. I have raised the contribution score from 2 to 3.

---

> > > ### Author Response · Authors · 2024-08-11
> > >
> > > Thank you for increasing the contribution score. We appreciate this positive change! However, for clarification, we would like to note that in typical NeurIPS review processes, reviewers often adjust the overall rating rather than the contribution score alone.

---

> > ### Author Response · Authors · 2024-08-14
> >
> > Dear Reviewer smsh,
> >
> > May we kindly ask if you would consider revising the overall rating based on our responses?
> >
> > Sincerely,
> >
> > The Authors

---

### Official Review · Reviewer_mRZs · 2024-07-15

**Soundness:** 4
**Presentation:** 3
**Contribution:** 3
**Rating:** 7
**Confidence:** 4

**Summary:**

The paper proposes the a novel framework called DRAGON, which uses a learnable probability distribution over a range of real numbers for the fractional order in graph dynamics, generalizing previous continuous GNN models. The paper provides a non-Markovian graph random walk interpretation for the DRAGON framework, assuming the feature updating dynamics adheres to a diffusion principle. It also proves that the DRAGON framework can approximate any waiting time distribution for graph random walks. The paper integrates DRAGON into several existing continuous GNN models and conducts experiments on various graph benchmarks. The results show that the DRAGON framework outperforms other methods on long-range graph datasets and improves the performance of continuous backbones on homophilic and heterophilic datasets.

**Strengths:**

1. The paper introduces the DRAGON framework, which incorporates distributed-order fractional calculus into continuous GNNs, a novel approach that goes beyond the traditional integer-order or single fractional-order differential equations used in previous GNN models.
The use of a learnable measure over a range of real numbers for the fractional order is a unique feature of DRAGON, which allows for a flexible superposition of multiple derivative orders and captures complex graph feature updating dynamics beyond the reach of conventional models.

2. The paper is clearly written and well-structured. The paper provides a non-Markovian graph random walk interpretation for DRAGON, which is a new perspective in understanding the dynamics of graph neural networks. The experimental results are presented in a clear and organized way, with tables and figures that help readers easily understand the performance of the DRAGON framework compared to other methods.

**Weaknesses:**

1. The DRAGON framework involves distributed-order fractional calculus and a learnable measure, which may increase the complexity of the model and make it more difficult to understand and implement. This complexity could potentially limit its adoption in practical applications.

2. The paper mainly focuses on a few datasets for evaluation, and the experiments could be further expanded to include more diverse and challenging datasets.

**Questions:**

see the above weaknesses

**Limitations:**

the author has adequately addressed the limitations in this paper.

---

> ### Author Rebuttal · Authors · 2024-08-07
>
> ## Weakness 1: Implementation and complexity details
>
> **Response:**  Thank you for your insightful comments. We are pleased to provide additional explanation regarding the implementation and complexity of our framework.
>
> Solving the distributed-order FDE Eq.(10) consists of two steps:
>
> - **Step 1:** Discretizing the distributed-order derivative by classical quadrature rule.
>     For example, suppose $w(\alpha)=\mu'(\alpha)$, applying composite Trapezoid rule [2-3] gives
>    \begin{align*}
>            \displaystyle\int_a^b D^\alpha X(t)d\mu(\alpha) = \frac{\Delta \alpha}{2} \left[w(\alpha_0) D^{\alpha_0}X(t) + 2\sum_{j=1}^{n-1} w(\alpha_j)D^{\alpha_j} X(t) + w(\alpha_n)D^{\alpha_n}X(t)\right]+ O((\Delta\alpha)^2),   \quad  \quad  \quad   \quad \quad \text{(S1)}
>    \end{align*}
>     where $\Delta\alpha=(b-a)/n$ and $\alpha_j=a+j\Delta\alpha$. After omitting small terms, we get the multi-term fractional differential equation Eq.(14).
>
> - **Step 2:** Solving  the multi-term fractional differential equation Eq.(14) using fractional Adams–Bashforth–Moulton method Eq.(28) or Grünwald-Letnikov method Eq.(36).
>
> For additional details on **error analysis**, we kindly direct the reviewer to **our Response to Weaknesses 2 for Reviewer uHnh**. For the **model complexity**, please refer to the top **global response section**.
>
> We hope this explanation alleviates concerns regarding the complexity and illustrates the DRAGON framework's potential for broad applicability.
>
> ## Weakness 2:  More diverse and challenging datasets
>
> **Response:** Thank you very much for your thoughtful suggestion. We value the chance to further elucidate the aims and outcomes of our study. Our research demonstrates how the DRAGON framework can enhance various continuous GNN models by integrating distributed-order fractional derivatives, thus boosting performance. Our experimental setup follows the methods outlined in the continuous GNNs literature.
>
> Additionally, in the manuscript, we have broadened our tests to include the Long Range Graph Benchmark, which features datasets such as Chemistry, to underscore the unique benefits of our DRAGON.
>
> The broad applicability of the DRAGON framework is notable, with potential uses in fields such as traffic forecasting [1] and learning system dynamics [2]. In the rebuttal, we have adapted integer-order differential equations in [1] to distributed-order fractional derivatives, expanding our framework's reach. Specifically, we have applied the DRAGON framework to the STG-NCDE model in preliminary time-series traffic forecasting tests, as demonstrated in Table R4. This preliminary result shows our framework's adaptability and effectiveness even without extensive parameter tuning.
>
> [1] Choi J, et al. Graph neural controlled differential equations for traffic forecasting. AAAI 2022
>
> [2] Huang Z, et al. Coupled graph ODE for learning interacting system dynamics. KDD 2021.
>
> **Table R4: Forecasting error(%) for traffic time-series data**
>
> |         | PeMSD4  | PeMSD4 | PeMSD4 | PeMSD7 | PeMSD7 | PeMSD7 |
> |--------------|------------|-------------|-------------|------------|-------------|-------------|
> | Model        | MAE | RMSE | MAPE | MAE | RMSE | MAPE |
> | STGODE       | 20.84      | 32.82       | 13.77       | 22.59      | 37.54       | 10.14       |
> | STG-NCDE     | 19.21      | 31.09       | 12.76       | 20.53      | 33.84       | 8.80        |
> | D-STG-NCDE   | 19.05      | 31.03       | 12.49       | 20.26      | 33.29       | 8.39        |

---

> ### Author Response · Authors · 2024-08-12
>
> Dear Reviewer mRZs,
>
> We sincerely appreciate the time and expertise you have dedicated to reviewing our paper. We deeply value your support.
>
> May we kindly ask for your confirmation on whether our rebuttal has adequately addressed your concerns? Your feedback is crucial to us.
>
> Thank you once again for your invaluable contributions and for sharing your expertise.
>
> With sincere gratitude,
>
> The Authors

---

### Official Review · Reviewer_uHnh · 2024-07-17

**Soundness:** 3
**Presentation:** 3
**Contribution:** 3
**Rating:** 6
**Confidence:** 3

**Summary:**

This paper presented a new type of continuous GNN that extends and unified many continuous GNN variants. The paper mainly generalize the distributed-order fractional derivatives to continuous GNN dynamics, and it now supports a mixing of fractional derivatives within a range of continuous orders. The author also presents many explanations for the presented generalization of continuous GNNs, with theoretical support and connections to non-markov random walk. Extensive experiments that coverage node-level and graph-level tasks are conducted, showing consistent improvement over existing baselines.

**Strengths:**

1. The presented framework that relies on distributed-order fractional derivatives generalizes many existing methods in the continuous GNN area, providing good theoretical support and insight for future research.
2. While some terms and math are not familiar to me, the author present the framework and main designs greatly and easy to follow. Motivations and theorem conclusions are good to understand.
3. Extensive experiments are conducted, and the performance improvement is notable and consistent.

**Weaknesses:**

1. While the author has conducted extensive experiments covering various domains, the hyperparameter selection strategy is not mentioned. Given the framework introduces many additional hyperparameters, the author should clearly state how they choose all hyperparameters, whether it is solely based on validation dataset.
2. The author mentioned how to solve DRAGON that contains the distributed-order fractional differential equation. It seems that the final computation equation is just the equation 36 mentioned in appendix. As the equation contains approximations, can the author clearly state the approximation error to the true solution? It seems that the equation 36 is a simple iterative based equation, and I'm not sure whether solving the distributed-order differential equation is that simple.
3. The complexity analysis is not very detailed. C and E is kind of vague.

**Questions:**

Except the questions listed above, can the author provide more computational complexity report? Table 8 and 9 only studies cora, I would like to see the report on large datasets. (large graph & large number of graphs)

**Limitations:**

The author has clearly stated.

---

> ### Author Rebuttal · Authors · 2024-08-07
>
> ## Weakness 1: Hyperparameters chosen
>
> **Response:** We appreciate the reviewer's comments.
>
> Like GRAND, CDE, and FROND, we employ a grid search on the validation dataset to optimize common hyperparameters such as hidden dimensions, learning rate, weight decay, and dropout rate.
> Some details are in Table 18 in our manuscript.
>
> Here we also clarify the parameters $\alpha_j$ in our implementation, distinct from other continuous GNNs.
> As detailed in Sec. 4.1, we restrict $\alpha_j$ to $[0,1]$, with values evenly spaced.
> We consistently apply the same 10 levels of $\alpha_j$, from 0.1 to 1.0, across all datasets, each associated with learnable weight $w_j$.  Our approach, therefore, avoids the need to extensively fine-tune the derivative order $\alpha$, unlike the FROND model. Furthermore, in Sec. G6, we demonstrate that this discretization is not sensitive to the number of $\alpha_j$ levels used.
>
> ## Weakness 2: Numerical solver and the approximation error
>
> **Response:** Solving the distributed-order FDE Eq.(10) consists of two steps:
>
> - **Step 1:** Discretizing the distributed-order derivative by classical quadrature rule.
>     For example, suppose $w(\alpha)=\mu'(\alpha)$, applying composite Trapezoid rule [2-3] gives
>    \begin{align*}
>            \displaystyle\int_a^b D^\alpha X(t)d\mu(\alpha) = \frac{\Delta \alpha}{2} \left[w(\alpha_0) D^{\alpha_0}X(t) + 2\sum_{j=1}^{n-1} w(\alpha_j)D^{\alpha_j} X(t) + w(\alpha_n)D^{\alpha_n}X(t)\right]+ O((\Delta\alpha)^2),   \quad  \quad  \quad   \quad \quad \text{(S1)}
>    \end{align*}
>     where $\Delta\alpha=(b-a)/n$ and $\alpha_j=a+j\Delta\alpha$. After omitting small terms, we get the multi-term fractional differential equation Eq.(14).
>
> - **Step 2:** Solving the  multi-term fractional differential equation Eq.(14) using fractional Adams–Bashforth–Moulton method Eq.(28) or Grünwald-Letnikov method Eq.(36).
>
>
> __Therefore, the approximation error to the true solution consists of numerical quadrature error in Step 1 and numerical solver error in Step 2.__ The former one is clear from (S1). For the latter, we address the main idea by considering the general $n^{th}$ order multi-term fractional differential equation: $\sum_{j=1}^n w_j D^{\alpha_j} y(t) = f(t) $ with initial condition $y(0) = y_0$. We have
>
> > **1).** For fractional Adams–Bashforth–Moulton method Eq.(28), the multi-term fractional differential equations are equivalently transformed into a system of single-term equations, which is then addressed using the solver for single terms as detailed in Eq.(28). The approximation error for this solution is quantified as follows [1]:
> \begin{align*}
>     \max_{j=0,1,\ldots,N} \left| y(t_j) - y_j \right| = O(h^{1+\min\{\alpha_j\}}),  \quad  \quad \quad  \quad \quad   \quad \quad \text{(S2)}
> \end{align*}
> where $y_j$ denotes the value of the solution at time $t_j$ as computed by the numerical method, and $y(t_j)$ represents the exact solution at time $t_j$, $h$ is the step size.
>
> > **2).** For Grünwald-Letnikov method Eq.(36), we apply Grünwald-Letnikov approximation [4] for each fractional derivative $D^{\alpha_j} y(t)$ that is given by:
> \begin{align*}
>     D^{\alpha_j} y(t_i) = \frac{1}{h^{\alpha_j}} \sum_{k=0}^{i} (-1)^k \binom{\alpha_j}{k} [y(t_{i-k}) - y_0] +O(h).
> \end{align*}
> Using techniques in [5], we get the approximation error below
> \begin{align*}
> \max_{j=0,1,\ldots,N} \left| y(t_j) - y_j \right| = O(h). \quad  \quad  \quad   \quad \quad \text{(S3)}
> \end{align*}
>
> The total error $E_{\text{total}}$ is actually a combination of approximation error in Step 1 and Step 2.
>
> [1].Diethelm K, et al. Detailed error analysis for a fractional Adams method.
>
> [2].Gao G, et al. Two alternating direction implicit difference schemes for two-dimensional distributed-order fractional diffusion equations.
>
> [3].Quarteroni A, et al. Numerical mathematics.
>
> [4].Podlubny I. Fractional differential equations Elsevier, 1998.
>
> [5].Jin B, et al. Correction of high-order BDF convolution quadrature for fractional evolution equations
>
> ## Weakness 3: Complexity analysis details
>
> **Response:**  Thank you for your feedback. We clarify here the notation and detail the computational complexity for our DRAGON using the above two numerical solvers.
>
>
> > The term $E = \frac{T}{h}$ quantifies the discretization (iteration) steps necessary for the integration process. Here, $T$ represents the integration time, $h$ is the step size, and $E$ denotes the total number of iterations required.
>
> > The term $C$ denotes the computational complexity of function $\mathcal{F}$. For instance, setting $\mathcal{F}$ to the GRAND model for evaluating the soft adjacency matrix results in $C =O( |\mathcal{E}| d)$, where $|\mathcal{E}|$ represents the edge set size and $d$ the dimensionality of the features (cf. [6]). Alternatively, using the GREAD model results in $C=O((|\mathcal{E}| + |\mathcal{E}_2|)d + |\mathcal{E}| d\_{\text{max}})$, where $|\mathcal{E}\_2|$ counts the two-hop edges, and $d\_{\text{max}}$ is the maximum degree among nodes (cf. [7]).
>
> For a detailed analysis of the **model complexity**, please refer to the top **global response section.**
>
> [6] Chamberlain, Ben, et al. "Grand: Graph neural diffusion." ICML, 2021.
>
> [7] Choi, Jeongwhan, et al. "Gread: Graph neural reaction-diffusion networks." ICML 2023.
>
> ## Question 1: More computational complexity report
>
> We follow the suggestions and prove the computational complexity on large datasets including Ogbn-arxiv as well as on a large number of graphs within the Peptides-func and Peptides-struct datasets. The results of this analysis are detailed in **Table R1, R2, and R3** in the top **global response section**. These results demonstrate that while our framework slightly increases computational costs compared to baseline continuous GNN models, it remains feasible for large graph datasets applications.

---

> > ### Comment · Reviewer_uHnh · 2024-08-10
> >
> > Thank you for these detailed response and additional measurement. Your algorithm seems to be well designed: with both theoretical support and limited computational overhead. Based on (s2) and (s3), can you give some insight to the measurement of the real error in experiments? Like how the error changes and affect the performance with respect to h in real experiments.

---

> > > ### Author Response · Authors · 2024-08-13
> > >
> > > Dear Reviewer uHnh,
> > >
> > > Thank you for your insightful questions, which have contributed to the enhancement of our paper.
> > >
> > > Could you please let us know if our responses have addressed your concerns adequately? We appreciate your guidance and look forward to your feedback.
> > >
> > > With sincere gratitude,
> > >
> > > The Authors

---

> > > ### Author Response · Authors · 2024-08-14
> > >
> > > Dear Reviewer uHnh,
> > >
> > > Thank you for your insightful questions, which have contributed to improving our manuscript. As the rebuttal period is drawing to a close, we kindly request your feedback on whether our responses have sufficiently addressed your concerns. We value your guidance and look forward to your further comments.
> > >
> > > Sincerely,
> > >
> > > The Authors

---

> ### Author Response · Authors · 2024-08-12
>
> Thank you for the new feedback!
>
> We would first like to clarify potential ambiguities surrounding the term "error" by distinguishing between "performance error" and "numerical error":
>
> > Performance error refers to the efficacy of GNNs in tasks such as node classification, where the focus is on the model's ability to correctly predict outcomes.
>
> > Numerical error, on the other hand, concerns the accuracy of numerical solutions to fractional differential equations compared to the "true" solution trajectory, which is a key issue in computational mathematics.
>
> The two errors are related, but not fully equivalent to each other.  The approximation error in (S2) and (S3) refers specifically to the "numerical error."
> For further clarity, we then conducted ablation studies on the Cora dataset to observe how errors change with respect to the step size $h$.
>
>
> > (1). In the first ablation study, we fixed all other parameters and varied the step size $h$ used in each experiment. We trained the model using different step sizes $h$ during the training phase and tested it with the corresponding step sizes during the test phase. The results are presented in **Table R5**. According to (S2) and (S3), the numerical approximation error should be large when $h$ is large. However, the results from Table R5 indicate that while classification performance deteriorates with larger step size $h$, it does not degrade to an unreasonable level and still maintains adequate classification performance. This occurs because both the training and testing phases follow the same discretization procedure, and although both are far from the true FDE solution, the loss function is designed to minimize the final classification error, resulting in satisfactory performance.
>
> > (2). In the second ablation study, we maintained fixed parameters as in the first study but changed our approach by training the model exclusively with a small step size $h=0.1$. During the testing phase, we varied the step size $h$. The results are presented in **Table R6**. According to (S2) and (S3), the numerical approximation error should be minimal when $h$ is small, and the solution with $h=0.1$ can be presumed close to the "true" solution trajectory of the FDE.
> We noted that when $h < 1$ and remains relatively small, the model still achieves good classification performance. This occurs because the "numerical error" is still minimal, keeping the numerical solution close to the "true" solution trajectory.
> However, as $h$ increases beyond this range, the "numerical error" grows significantly, diverging from both the "true" solution trajectory and the approximate solution with $h=0.1$. Consequently, classification performance deteriorates substantially.
>
> **Table R5: Step size and classification accuracy(\%) (training and test)**
>
> | Step size | 5     | 2     | 1     | 0.5   | 0.2   | 0.1   |
> |-------------|-------|-------|-------|-------|-------|-------|
> | Solver Eq.(28)   | 80.24 | 82.98 | 83.11 | 83.18 | 83.15 | 83.11 |
> | Solver Eq.(36)    | 80.03 | 82.33 | 82.57 | 82.91 | 83.11 | 83.19 |
>
>
> **Table R6:  Step size and classification accuracy(\%) (test phase)**
>
> | Step size  | 5     | 2     | 1     | 0.5   | 0.2   | 0.1   |
> |------------|-------|-------|-------|-------|-------|-------|
> | Solver Eq.(28)   | 35.23 | 65.48 | 74.42 | 80.91 | 81.02 | 83.11 |
> | Solver Eq.(36)   | 39.80 | 61.62 | 73.91 | 76.14 | 80.41 | 83.19 |

---

### Author Rebuttal · Authors · 2024-08-07

We sincerely thank all the reviewers for their insightful comments and valuable suggestions. We greatly appreciate the feedback and have thoughtfully addressed each point in our detailed responses.

In this "global" response, we provide a detailed analysis of the computational complexity of the DRAGON framework as suggested by reviewers uHnh, mRZs, and smsh. Additionally, we include further numerical results that substantiate our responses.

Moreover, in the attached PDF file, we present visualizations of the non-Markovian graph random walk, as recommended by Reviewer smsh.

**Model Complexity**
In our manuscript, the time complexity of the DRAGON framework is discussed in Section 4.4. Here, we provide a more detailed analysis concerning the two numerical solvers employed within our study.

For the Adams–Bashforth–Moulton method (Equation 28), we compute $\mathbf{X}\_{k+1} = \mathbf{X}\_0 + \frac{1}{\Gamma(\alpha)} \sum_{j=0}^{k} b_{j, k+1} \mathcal{F}(\mathbf{W}, \mathbf{X}\_j)$. This process necessitates repeated computation of $\mathcal{F}(\mathbf{W}, \mathbf{X}\_j)$ at each iteration. Direct computation leads to a complexity of $\mathcal{O}(C E^2)$. If we save the
intermedia function evaluation values $\\{\mathcal{F}(\mathbf{W}, \mathbf{X}\_j)\\}\_j$, the total computational complexity over the entire process can be expressed as $\sum_{k=0}^E (C + O(k))$, where $O(k)$ represents the computational overhead of summing and weighting the $k$ terms at each step. We adopt this strategy in our code implementation. If the cost of weighted summing is minimal, the complexity reduces to $O(E C)$. For the Grünwald-Letnikov method (Equation 36), the computational complexity is $O(EC)$ since it requires no repeated computation of $\mathcal{F}(\mathbf{W}, \mathbf{X}_j)$, with only one $\mathcal{F}$ computation per iteration.

> The term $E = \frac{T}{h}$ quantifies the discretization (iteration) steps necessary for the integration process. Here, $T$ represents the integration time, $h$ is the step size, and $E$ denotes the total number of iterations required.

> The term $C$ denotes the computational complexity of function $\mathcal{F}$. For instance, setting $\mathcal{F}$ to the GRAND model for evaluating the soft adjacency matrix results in $C = |\mathcal{E}| d$, where $|\mathcal{E}|$ represents the edge set size and $d$ the dimensionality of the features (cf. [6]). Alternatively, using the GREAD model results in $C=O((|\mathcal{E}| + |\mathcal{E}_2|)d + |\mathcal{E}| d\_{\text{max}})$, where $|\mathcal{E}\_2|$ counts the two-hop edges, and $d\_{\text{max}}$ is the maximum degree among nodes (cf. [7]).

This refined complexity analysis will be included in the revised paper version.

[6] Chamberlain, Ben, et al. "Grand: Graph neural diffusion." ICML, 2021.

[7] Choi, Jeongwhan, et al. "Gread: Graph neural reaction-diffusion networks." ICML 2023.

**Table R1:** Computation time of models on the Ogbn-arxiv dataset

| Model         | D-GRAND-l | D-GRAND-nl | D-GraphCON-l | D-GraphCON-nl |
|---------------|-----------|------------|--------------|---------------|
| Inf. Time(s)  | 0.083     | 0.139      | 0.141        | 0.196         |
| Train. Time(s)| 0.33      | 0.67       | 0.57         | 0.92         |


| Model         | F-GRAND-l | F-GRAND-nl | F-GraphCON-l | F-GraphCON-nl |
|---------------|-----------|------------|--------------|---------------|
| Inf. Time(s)  | 0.047     | 0.108      | 0.062        | 0.123         |
| Train. Time(s)| 0.14      | 0.53       | 0.50         | 0.59         |

| Model         | GRAND-l   | GRAND-nl   | GraphCON-l   | GraphCON-nl   |
|---------------|-----------|------------|--------------|---------------|
| Inf. Time(s)  | 0.038     | 0.099      | 0.044        | 0.105         |
| Train. Time(s)| 0.10      | 0.51       | 0.15         | 0.55           |

***

**Table R2:** Computation time of models on the Peptides-func dataset

| Model         | D-GRAND-l | F-GRAND-l | GRAND-l |
|---------------|-----------|-----------|---------|
| Inf. Time(s)  | 0.324     | 0.275     | 0.259   |
| Train. Time(s)| 2.853     | 2.298     | 2.034   |

***

**Table R3:** Computation time of models on the Peptides-struct dataset

| Model         | D-GRAND-l | F-GRAND-l | GRAND-l |
|---------------|-----------|-----------|---------|
| Inf. Time(s)  | 0.338     | 0.269     | 0.253   |
| Train. Time(s)| 2.923     | 2.258     | 2.008   |

---

### Decision · Program_Chairs · 2024-09-25

**Decision:**

Accept (spotlight)

**Comment:**

This paper presented DRAGON, a new type of continuous GNN that extends and unifies many continuous GNN variants.
DRAGON provides a non-Markovian graph random walk interpretation. Unlike traditional continuous GNNs that utilize integer-order or single fractional-order differential equations, DRAGON employs a learnable probability distribution over a range of real numbers for the derivative orders.

The reviews are quite enthusiastic and the responses of the authors are quite solid.